# Phage proteins target and co-opt host ribosomes immediately upon infection

**Milan Gerovac** [1,2], **Kotaro Chihara** [2], **Laura Wicke** [3], **Bettina Böttcher** [4], **Rob Lavigne** [3] **& Jörg Vogel** [1,2] ✉

Bacteriophages must seize control of the host gene expression machinery to replicate. To bypass bacterial anti-phage defence systems, this host takeover occurs immediately upon infection. A general understanding of phage mechanisms for immediate targeting of host transcription and translation processes is lacking. Here we introduce an integrative high-throughput approach to uncover phage-encoded proteins that target the gene expression machinery of *Pseudomonas aeruginosa* immediately upon infection with the jumbo phage ΦKZ. By integrating biochemical, genetic and structural analyses, we identify an abundant and conserved phage factor ΦKZ014 that targets the large ribosomal subunit by binding the 5S ribosomal RNA, and rapidly promotes replication in several clinical isolates. ΦKZ014 is among the earliest ΦKZ proteins expressed after infection and remains bound to ribosomes during the entire translation cycle. Our study provides a strategy to decipher molecular components of phage-mediated host takeover and argues that phage genomes represent an untapped discovery space for proteins that modulate the host gene expression machinery.

Bacteriophages hold great promise for use in biotechnological applications and antibacterial design strategies because they modify fundamental processes of life to subvert their bacterial host. Jumbo phages present particularly exciting opportunities for identifying molecular factors with functions for a successful host takeover because they feature extraordinarily large genomes (200–600 kb) that encode hundreds of proteins with currently unknown functions. One hallmark model is ΦKZ, a jumbo phage that infects *Pseudomonas aeruginosa*, a Gram-negative bacterium and a major cause of nosocomial and antibiotic-resistant infections[1,2]. ΦKZ shares a strikingly complex infection cycle with other *Chimalliviridae* jumbo phages[3], which involves the formation of a phage nucleus that spatially separates genome replication and transcription from translation in the cytosol[4]. The phage nucleus excludes DNA-targeting host defence systems, rendering the phage resistant to certain CRISPR–Cas systems[5,6].

Most phages require mechanisms that enable them to immediately take control of gene expression in the infected cell to efficiently proceed through the replication process. It is well established that they manipulate the host RNA polymerase (RNAP) and overload the translation machinery with phage-derived transcripts. Phages encode alternative σ-factors for selective initiation on phage transcripts, factors that modulate different stages of the transcription cycle, or enzymes that modify RNAP residues by ADP ribosylation and phosphorylation (reviewed in ref. 7). In addition, phages can encode their own single or multi-subunit RNAPs[8,9]. For example, ΦKZ encodes specialized RNAPs[10,11] but also an inhibitor of the major host endoribonuclease[12]. In contrast to the wealth of phage factors that modulate host transcription, host takeover at the level of translation is understudied[13]. Although previous studies provided evidence that ribosome-associated phage factors might exist[14,15], a physical interaction with ribosomes was never shown. Of note, rapid translation of phage proteins is also imperative to overcome stress responses and phage defence systems in the host that act immediately after phage infection[16]. Examples of phage-derived counter-defences are anti-restriction-modification proteins, anti-CRISPR proteins and

[1]Institute for Molecular Infection Biology (IMIB), University of Würzburg, Würzburg, Germany. [2]Helmholtz Institute for RNA-based Infection Research (HIRI), Helmholtz Centre for Infection Research (HZI), Würzburg, Germany. [3]Laboratory of Gene Technology, KU Leuven, Leuven, Belgium. [4]Biocenter and Rudolf Virchow Center for Integrative and Translational Bioimaging, University of Würzburg, Würzburg, Germany. ✉e-mail: joerg.vogel@uni-wuerzburg.de

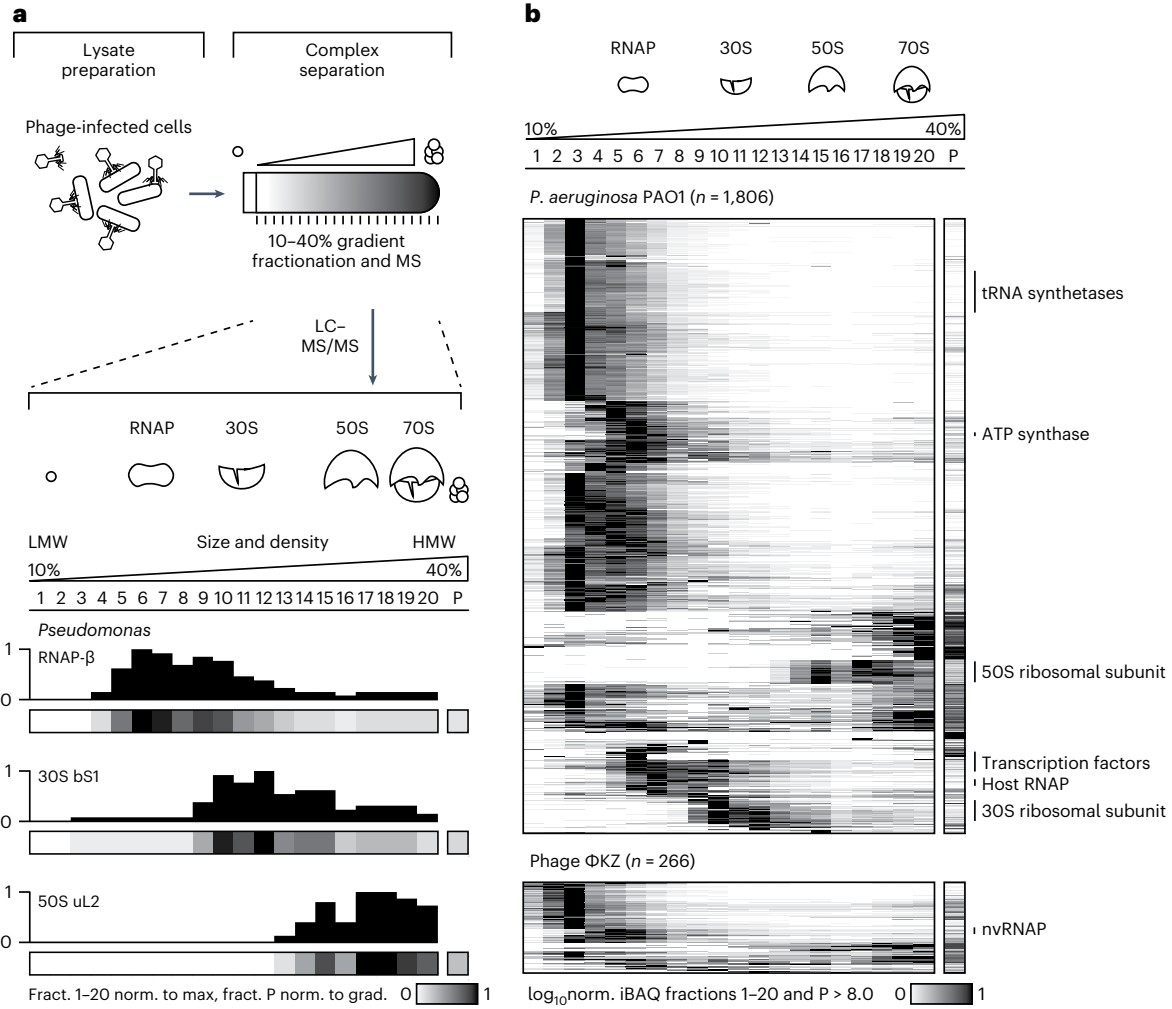

**Fig. 1 | ΦKZ proteins sediment in HMW fractions. a**, Cellular lysates of *P. aeruginosa* cells (strain PAO1) infected with ΦKZ were separated in a glycerol gradient by size and density into 20 fractions plus the pellet. Mass spectrometric quantification yielded sedimentation profiles of individual proteins, for example, the RNAP β-subunit, 30S ribosomal protein bS1 and 50S ribosomal protein uL2. For quantification, fractions 1–20 are normalized (norm.) to the fraction (fract.) with the maximum protein abundance (set to 1, black) and the pellet fraction P is normalized to the protein abundance in the entire gradient (grad.) (sum equals 1). **b**, Sedimentation profiles of *P. aeruginosa* (top) or ΦKZ (bottom) proteins were hierarchically clustered. Host proteins in HMW fractions correspond to subunits of the gene expression machinery. ΦKZ viral polymerase subunits sedimented in fraction five. The data are a representative example of two independent experiments. Protein abundance relative $\log_{10}$ iBAQ > 8.0. LC, liquid chromatography.

anti-toxin proteins[17,18]. Still, we have only preliminary knowledge of the true number and temporal activity of phage-encoded factors, especially those that act in the first line of host takeover.

In this study, we undertook a systematic analysis of cellular protein complexes in ΦKZ-infected *Pseudomonas* cells to identify phage factors that engage with the host gene expression machinery early during infection. We observed several proteins that form or associate with large complexes or co-sediment with phage or host RNAPs or ribosomes. The molecular characterization of one of these phage factors shows that it acts during early infection and targets a core region of the bacterial protein synthesis machinery during host takeover.

## Results

### Phage factors fractionate with the gene expression machinery

To systematically identify phage proteins that interact with and subvert the host gene expression machinery, we used gradient fractionation coupled with sequencing and proteomics (Grad-seq[19]). This method predicts molecular complexes based on the separation of cellular lysates by a classical glycerol gradient, followed by high-throughput RNA sequencing (RNA-seq) and mass spectrometry (MS) analyses of the

individual gradient fractions. Our previous work established that major *P. aeruginosa* protein complexes remain intact upon phage infection[20]. Here, to capture early host takeover events, we infected *P. aeruginosa* with ΦKZ for 10 min, 20 min and 30 min and analysed pooled samples in a 10–40% glycerol gradient, which is most powerful in resolving complexes between >200 kDa and 4 MDa. A total of 20 gradient fractions were then subjected to MS for proteomic analysis. As previously described, subunits of large metabolic complexes, the host RNAP and ribosomes can be identified based on correlated sedimentation profiles[20] (Fig. 1a and Extended Data Fig. 1a). After infection, the apparent bacterial proteome appears intact on denaturing gels (Extended Data Fig. 1b). Quantitative MS analysis identified ~10% of the total protein count as being phage derived (Extended Data Fig. 1c).

Our sampling approach enabled us to detect almost three-quarters (266 of 369) of the annotated ΦKZ proteins. These data also serve to validate that these phage genes indeed produce proteins (at relative $\log_{10}$ intensity-based absolute quantification (iBAQ) > 8; Fig. 1b and Supplementary Table 1). Similar to host proteins, most phage proteins are present in the first 3 low-molecular-weight (LMW) fractions, suggesting they are monomeric or form small <100 kDa complexes. Nevertheless,

~30 phage proteins are present in high-molecular-weight (HMW) fractions (Fig. 1b). This includes six annotated structural virion proteins that also accumulated in the pellet fraction, indicating they are probably part of virion assembly intermediates (Fig. 2a). In addition, helicase ΦKZ203, DNA-binding protein ΦKZ207, nucleases ΦKZ056/-179 and ATPases ΦKZ208/-286 sedimented in HMW fractions, suggesting they are part of larger complexes with as-yet-uncharacterized host or phage factors (Fig. 2a). Moreover, we find many functionally uncharacterized phage factors that sediment in HMW fractions, providing a rich resource for further exploration that is supported by a newly programmed interactive online explorer (www.helmholtz-hiri.de/en/datasets/gradseqphage).

A global clustering analysis of highly correlated protein sedimentation profiles suggested the association of phage proteins with key gene expression complexes of *P. aeruginosa*, that is, RNAP and the small (30S) and large (50S) ribosomal subunits (Fig. 2b). For example, several phage proteins that have so far not been associated with transcription are present in RNAP fractions, suggesting their potential interaction with host RNAP (ΦKZ020, -056, -069, -297, -226 and -157). *Chimalliviridae* members inject a virion RNAP (vRNAP) with the phage genome for initial phage genome transcription, whereas a phage-encoded non-virion RNAP (nvRNAP) takes over transcription during later stages of infection in a host-RNAP-independent fashion[3,9,10]. We find that ΦKZ nvRNAP subunits (ΦKZ055, -068, -071, -073, -074 and -123)[10,11,21] and the vRNAP subunits ΦKZ080, -149, -178 and -180 cluster together, which further validates our clustering analysis. ΦKZ079, -122, -176 and -264 are also present in this cluster, suggesting that they are interaction partners of the nvRNAP. Notably, ΦKZ176 and -079 were previously shown to co-precipitate with the vRNAP subunit ΦKZ178 (ref. 10). Strikingly, we observed that nine phage proteins peak in ribosomal fractions but were absent from the pellet (Fig. 2), making them strong candidates for ribosome-targeting factors.

### Early expressed phage proteins form complexes with ribosomes

The 9 ΦKZ proteins observed in ribosomal fractions were predicted to target either the 30S (ΦKZ135 and -216) or the 50S subunit (ΦKZ014, -105, -110, -206, -225, -234.1 and -299) (Fig. 2 and Supplementary Table 1). Their function is currently unknown. To prioritize candidates that act on host protein synthesis immediately upon infection, we performed a high-resolution gene expression analysis, isolating total RNA of *P. aeruginosa* before and every 2 min after adding ΦKZ, followed by RNA-seq. We focussed on the first 10 min of the ~1 h ΦKZ infection cycle, which are crucial to establishing a productive infection. During this period, phage DNA is transcribed in the early phage infection vesicle[22] and host DNA is degraded[23].

Our RNA-seq time course revealed that phage mRNAs were expressed very rapidly and accounted for ~40% of all coding transcripts after 10 min (Fig. 3a and Supplementary Table 2). The most abundant transcripts included several that encode proteins present in ribosomal fractions, for example, ΦKZ014, -105 and -216 (Fig. 3b). Importantly, these mRNAs accumulated faster than the transcripts encoding nvRNAP subunits, the phage nucleus shell protein ChmA or the spindle-apparatus protein PhuZ, suggesting that ribosome targeting by ΦKZ during early host takeover is a high priority.

On the basis of these observations, we selected the proteins ΦKZ014 and ΦKZ105 to probe their interaction with translating ribosomes. We also included ΦKZ206, a protein that showed a very similar sedimentation profile to ΦKZ014 but whose expression peaked later. To confirm the sedimentation profile of these proteins, we ectopically expressed epitope-tagged ΦKZ014, -105 and -206 in uninfected *P. aeruginosa* and analysed their position in the glycerol gradient by immunoblotting. In agreement with the Grad-seq profiles, these proteins sedimented in ribosomal fractions (Fig. 3c). In contrast to native conditions, under which the proteins occurred solely in ribosomal

fractions (Fig. 2a), ΦKZ014 and ΦKZ105 were also present in LMW fractions, probably due to overexpression of the proteins. Overall these data confirm that all three phage proteins sediment in ribosomal fractions independent of infection and of other phage proteins. Moreover, ΦKZ014 and ΦKZ105 also accumulated in polysome fractions, suggesting a role in translating ribosomes (Fig. 3d).

### ΦKZ014 is abundant and targets the 50S ribosomal subunit

For further analysis, we focussed on ΦKZ014 because homologues of this protein are present in many ΦKZ-related phages (Phabio, SL2, ΦPA3, OMKO1, Psa21, EL, KTN4, PA1C, PA7, Fnug, vB_PaeM_PS119XW and PA02, based on PHROG identifier 16056 and PVOG identifier 9777; Fig. 4a), implying a conserved function. Indeed, the ΦKZ014 homologue gp099 of the phylogenetically distant *Pseudomonas* phage EL also co-fractionated with the 50S subunit (Fig. 4b).

Western blot analysis using a ΦKZ014 antibody and probing *P. aeruginosa* samples from an infection time course detected the protein as early as 5 min post-infection (Fig. 4c), in line with the early abundance of the ΦKZ014 mRNA (Fig. 3b). Protein expression plateaued after 10 min, at which stage the ΦKZ014 protein reached ~2,000 copies per cell. This copy number was inferred from a quantitative comparison of western blot signals for the protein in phage-infected cells with a standard provided by recombinantly produced ΦKZ014 protein that was purified under denaturing conditions (Extended Data Fig. 2).

Next we took two biochemical approaches to narrow down the interaction site of ΦKZ014 on the bacterial ribosome. First, we probed the ribosome interaction of ΦKZ014 at low-magnesium-ion conditions, when the 70S fraction dissociates into its two subunits. In these settings, ΦKZ014 shifts to the 50S fractions (Fig. 4d). At higher magnesium levels, which preserve 70S ribosomes, ΦKZ014 co-migrates with the 70S fractions. This confirms that ΦKZ014 targets the 50S subunit and that the interaction probably occurs outside of the inter-subunit site. Second, in an orthogonal protein pull-down assay using FLAG-tagged ΦKZ014 to identify candidate interaction partners of this phage protein in vivo after infection, we observed a strong enrichment of host proteins, especially uL30, bL35 and bL36, all of which are known constituents of the 50S ribosomal subunit (Fig. 4e).

Next we probed if the ribosome association of ΦKZ014 was mediated by interactions with mRNA, in which case it would be sensitive to treatment with micrococcal nuclease (MNase), or by the nascent protein chain, in which case it would be sensitive to release by the antibiotic puromycin. When transcripts were degraded by MNase, we observed the expected shift of the phage protein from the polysome fraction to 70S ribosomes, but there was no shift to LMW fractions (Fig. 4f). The sedimentation of ΦKZ014 was not altered upon puromycin treatment. Together, these observations confirm a stable and direct association of ΦKZ014 with host ribosomes independent of the transcript or nascent chain. In addition, we validated the association of native ΦKZ014 in infected cells with ribosomes (Fig. 4g,h).

### ΦKZ014 clamps the terminal end of 5S rRNA

To gain insights into the architecture of the ΦKZ014–ribosome complex, we expressed His-tagged ΦKZ014 in *Pseudomonas* and affinity purified ribosomes via the tagged protein (Fig. 5a). The complex was reconstructed by single-particle cryogenic electron microscopy (cryo-EM) (Fig. 5b, Extended Data Fig. 3 and Supplementary Table 3). Interestingly, we observed that ΦKZ014 occupied the 50S ribosomal subunit at the central protuberance (Fig. 5b,c), a region that facilitates communication between various functional ribosome sites[24]. The main constituent of the central protuberance is the 5S rRNA, which is essential for efficient translation[25,26]. Notably, we obtained 70S–ΦKZ014 maps with different tRNA occupancy, including the physiologically relevant aminoacyl (A) and peptidyl (P), and P and exit (E) sites (corresponding to the pre-classical and -hybrid 2* states, respectively)[27].

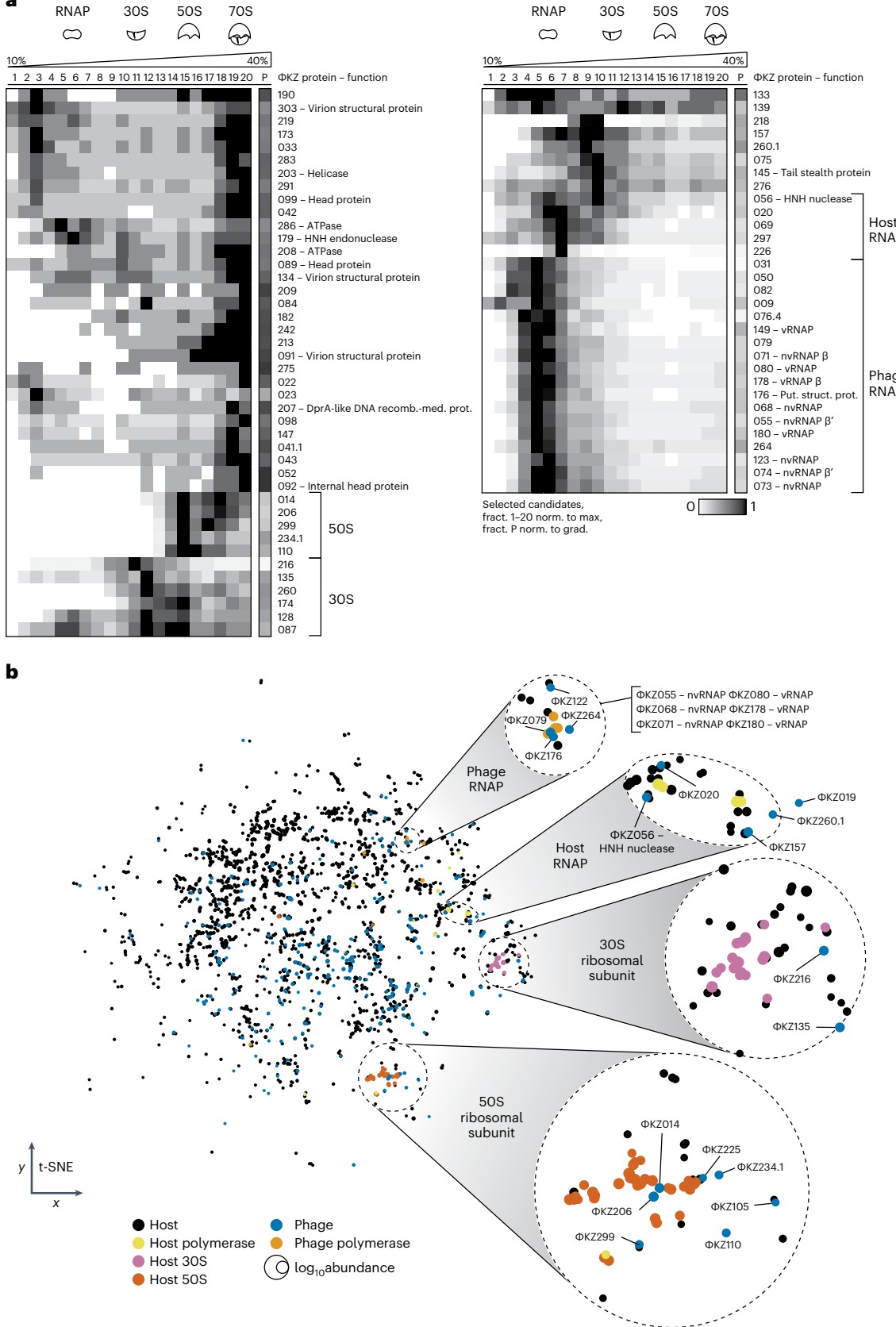

**Fig. 2 | ΦKZ proteins are co-sedimented with the host gene expression machinery. a**, Sedimentation profiles of selected ΦKZ proteins in HMW fractions. **b**, t-distributed stochastic neighbour embedding (t-SNE) clustering of sedimentation profiles revealed proteins that form large macromolecular complexes. ΦKZ RNAP subunits clustered together with ΦKZ079/-122/-176 and -264. ΦKZ019/-020/-56/-157 and -260.1 clustered with host RNAP proteins. ΦKZ216 and ΦKZ135 clustered with proteins of the 30S ribosomal subunit, and ΦKZ014/-105/-110/-206/-225/-234.1/-299 clustered with proteins of the 50S ribosomal subunit. Prot., protein; recomb.-med., recombination-mediator; put. struct., putative structural.

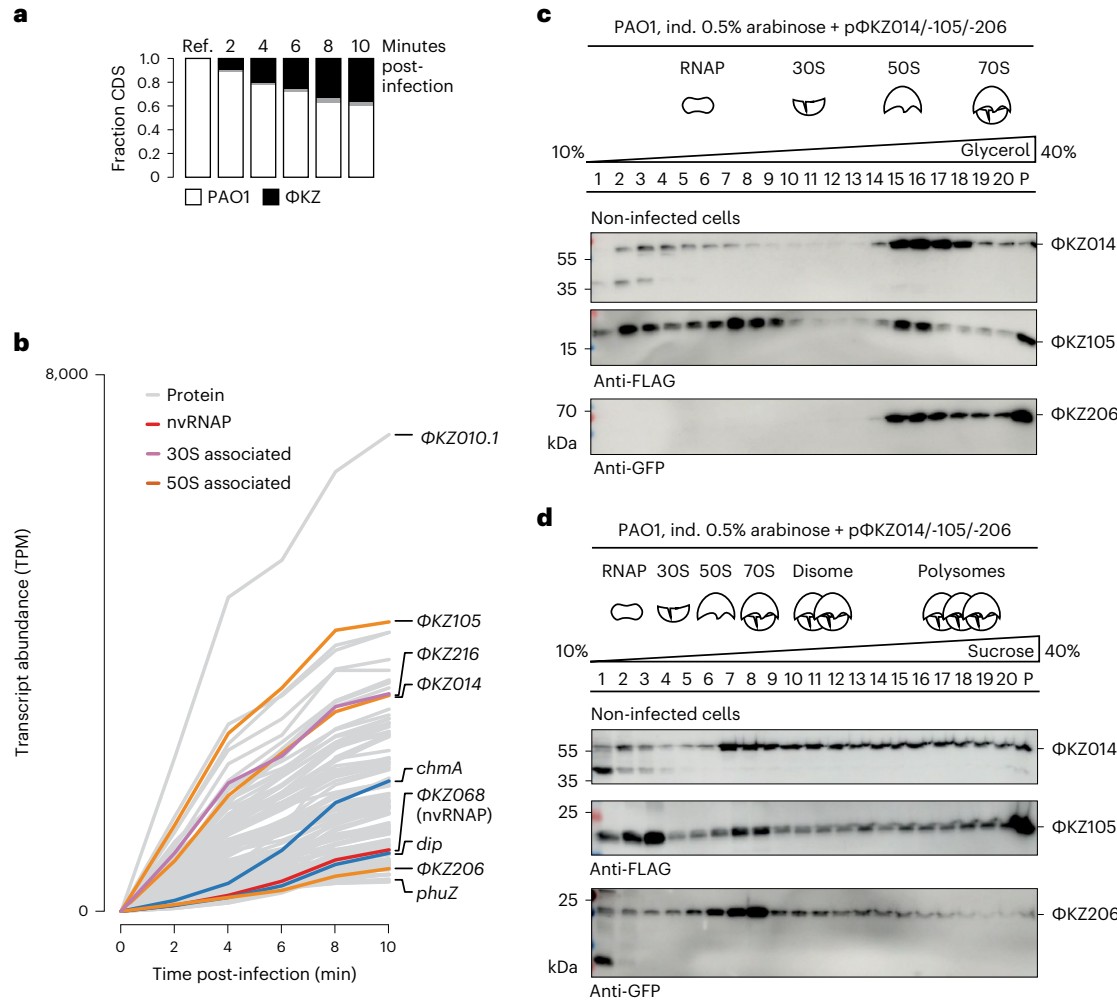

**Fig. 3 | Early expressed phage genes encode for factors that co-sediment with ribosomes. a**, *P. aeruginosa* (strain PAO1) was infected with ΦKZ and RNA was extracted and sequenced at the indicated time points. The fraction of PAO1 and ΦKZ coding sequences (CDS) is plotted. **b**, Phage transcript abundance over the first 10 min of infection based on RNA-seq analysis. Among the most abundant early transcripts, several encode proteins that sediment in ribosomal fractions (pink, 30S; orange, 50S). nvRNAP subunit ΦKZ068 (red), the RNA degradosome

inhibitor Dip and the phage nucleus protein ChmA are expressed at lower levels compared with the phage factors that are sedimented in ribosomal fractions. Cut-off at the sum of most abundant transcripts: >85% total phage reads; *n* = 115 transcripts. **c**,**d**, Cellular lysates of *Pseudomonas* cells that ectopically express epitope-tagged ΦKZ014, -105 and -206 were analysed in a glycerol gradient (**c**) or a sucrose gradient (**d**) followed by western blot analysis of the individual fractions. Ref., reference; ind., induced with.

Other states with tRNAs only in P and E sites were enriched, probably due to our purification conditions (Extended Data Fig. 3b).

ΦKZ014 consists of an amino-terminal (N-terminal) helix that is connected to a β-fold domain, followed by a central helical domain. Its predicted carboxy-terminal (C-terminal) helix–loop–helix domain is linked by a helix and is not resolved in the electron density map (Fig. 5c and Extended Data Fig. 4a). Strikingly, ΦKZ014 clamps the terminal end of 5S rRNA and interacts with it via charge-based interactions (Fig. 5d and Extended Data Fig. 4b). Notably, no factors were previously described to occupy the ribosome at the 5S rRNA site. The first uridine residue of the 5S rRNA is coordinated by the N-terminal ΦKZ014 residues Arg10, Lys15 and Trp18 together with the central domain residues Arg262 and Lys263. The second guanosine residue of 5S rRNA is coordinated by Arg23. This multivalent coordination leads to a flip of the uridine out of its helical conformation. ΦKZ014 is also in contact with the 23S rRNA, including the stem of the A-site finger (ASF) helix, which is important for the accommodation of tRNA in the A site. Two sequential histidines within a loop in the central domain (His178 and His179) infiltrate the 23S rRNA major groove at positions 842 and 908 (Extended Data Fig. 4c). This direct contact might lead to structural rearrangements in the ASF through rotational extension.

To validate the observed interaction sites with the 5S rRNA, we mutated to alanine several coordinating residues of ΦKZ014, that is, Lys15, Arg23 and Arg262. These coordinating residues are well conserved among various ΦKZ014 homologues (Extended Data Fig. 4d). These mutations (K15A, R23A, or both, and R262A) reduced the association of ΦKZ014 with ribosomes in 70S and polysome fractions, causing it to accumulate in LMW fractions (Fig. 5e). In addition, we observed reduced association for ΦKZ014 if Lys152, which interacts with A94 of the 5S rRNA, was exchanged for alanine (Extended Data Fig. 4c). Thus, our structural data validate the direct interaction of ΦKZ014 with the large ribosomal subunit and suggest that ΦKZ014 might play a role in translation via modulation of 5S rRNA or the stem of the ASF.

### ΦKZ014 plays a role in translation early during infection

To test if ΦKZ014 has broad effects on bacterial translation, we ectopically expressed the protein in uninfected host cells and pulse radiolabelled newly translated proteins with [³⁵S]methionine. Under these conditions, we did not observe altered host protein translation (Extended Data Fig. 5a). To probe the function of ΦKZ014 in the context of infection, we disrupted the gene in the phage genome (Extended Data Fig. 5b,c). The ΔΦKZ014 phage can be propagated in the laboratory

strain PAO1, indicating that the protein is not essential for phage replication in this strain (Extended Data Fig. 5d). We radiolabelled translated proteins at different time points during infection. In line with our previous observations (Figs. 3b and 4c), ΦKZ014 is immediately synthesized upon infection, similar to ChmA (Fig. 6a). Upon deletion of *ΦKZ014*, we observe several changes in the overall protein pattern, indicative of a modulatory role of ΦKZ014 on protein synthesis. This is also in agreement with the 70S–ΦKZ014 structural maps, which covered various translational states (Extended Data Fig. 3b). The impact of ΦKZ014 might become more apparent upon translational stress, which is not experienced by PAO1 under standard growth conditions.

To identify a host in which ΦKZ014 has a strong impact on phage replication, we screened a collection of clinical isolates of *P. aeruginosa* (PaLo1 to PaLo45) (Lood, C., unpublished; NCBI BioProject PRJNA731114) for altered plaque formation by ΔΦKZ014, compared with the wild-type (WT) phage. This identified *P. aeruginosa* strains in which the *ΦKZ014*-deletion phage caused reduced plaque counts or sizes (PaLo8 and PaLo39). In addition, we identified two strains (PaLo9 and PaLo44) in which ΔΦKZ014 did not yield plaques. These diverse phenotypes indicated pleiotropic effects on phage replication (Fig. 6b and Extended Data Fig. 5e). The most severe phenotype was apparent in PaLo44. The halo in the bacterial lawn is indicative of cell lysis. This observation is also supported by the fact that bacterial cells did not grow out at a high multiplicity of infection (MOI) of ΔΦKZ014 and that we observed a delayed outgrowth at low MOI (Fig. 6c). This phenotype is suggestive of the presence of a defence system, whose activation leads to abortive infection (Fig. 6b and Extended Data Fig. 5e). Ectopic expression of ΦKZ014 from a plasmid rescued plaque formation by the ΔΦKZ014 phage in the PaLo44 strain but did not induce more plaques upon infection with the WT phage. Hence ΦKZ014 is not rate limiting for phage replication in the PaLo44 strain. The successful complementation of ΦKZ014 activity by expression in trans rules out a potential polar effect of the ΔΦKZ014 mutation within the phage genome and shows that the protein can play an important role during phage replication in specific strains (Extended Data Fig. 5e).

To ensure that the ΔΦKZ014 phage was still able to infect the PaLo44 strain, we imaged bacterial cells at different time points after infection with either the WT or the ΔΦKZ014 phage. The injected phage DNA was visible at the cell pole immediately after infection in either case, hence initial infection is independent of the presence of ΦKZ014 (Fig. 6e). In cells infected by WT ΦKZ, the phage nucleus forms and is centred in the middle of the cell, as previously described[4]. In contrast, infection with the ΦKZ014-deficient phage halts and the injected phage genome remains associated with the cell pole.

The inability to form a phage nucleus may be linked to the lack of ChmA, the principal component of this nuclear shell. Upon infection with the WT phage, ChmA is apparent 15 min after infection in a Coomassie-stained gel of bacterial lysates, whereas cells infected with the ΔΦKZ014 phage fail to produce the ChmA protein (Fig. 6f). Pulse radiolabelling of newly translated proteins confirmed the lack of

ChmA production after infection with the ΔΦKZ014 phage and showed that production of this central protein for phage nucleus formation is blocked in the absence of *ΦKZ014* at a very early stage (Fig. 6g). In addition, we observe reduced de novo protein synthesis within the first 2.5 min upon infection with the ΔΦKZ014 phage, followed by a gradual shut-off over the course of 40 min. Therefore, we conclude that the ribosome-associated factor ΦKZ014 is essential for phage replication in strains that can abort phage replication via translational shutdown.

## Discussion

Jumbo phages, whose large genomes typically encode hundreds of functionally uncharacterized and evolutionarily untraceable proteins, represent an untapped reservoir of non-conserved protein families. However, the sheer number of these uncharacterized proteins requires experimental selection before functional analysis. The screening approach introduced here is applicable to diverse phages and can be used in the quest to systematically identify phage proteins that engage with host protein complexes, including the gene expression machinery. Coupling this analysis to gene expression profiling early during infection enabled us to select proteins that are likely to be important for immediate host takeover. A more detailed investigation of these candidates promises to yield insights into how the phage manipulates its host early during the infection process.

Whereas phages often provide their own RNAPs to ensure transcription of their genome, for protein synthesis they fully rely on the host translation machinery. This machinery comprises ribosomal RNAs and >50 protein components, and its size and complex assembly process makes it unlikely that a phage would be able to provide its own ribosomes, either physically or genetically. Thus, for a phage to seize control of protein synthesis, it must be able to modulate host ribosomes. Indeed, the initial prediction that phage proteins might associate with ribosomal subunits dates back >50 years[14,28], but the identity of such phage factors had remained elusive. Here we report several ΦKZ proteins that interact with the ribosome, as shown by ectopic expression of three of these proteins in uninfected *P. aeruginosa* cells (Fig. 3c,d). In contrast to phage-encoded proteins with sequence similarity to known ribosomal or translation factors that were predicted in metagenomic studies[29,30] or in phage λ[31], these ΦKZ proteins have no sequence similarity to any known host protein.

The discovery of phage proteins that modify host RNAP has been instrumental in understanding individual steps of the bacterial transcription cycle[7]. We envision a similar potential for ribosome-targeting phage proteins. For example, our structural data reveal that the ribosome-associated factor ΦKZ014 interacts with the 50S subunit in close contact with the 5S rRNA. No other proteins have been observed to interact at this site. As such, understanding how ΦKZ014 affects protein synthesis in the first minutes of infection may also help to unveil the molecular function of the universal 5S rRNA. Notably, more than half a century after the discovery of 5S rRNA[32], and despite its established importance for efficient translation[25] and during ribosome

---

**Fig. 4 | ΦKZ014 is conserved, is abundant and interacts directly with the large ribosomal subunit. a**, ΦKZ014 homologues were identified by PSI-BLAST in phage genomes. For comparative genome analysis, genomic loci were aligned to ΦKZ014 (red). Homologous genes were identified by sequence similarity and are colour coded. **b**, FLAG-tagged gp99 (gp99₋FLAG), the ΦKZ014 homologue in phage EL, was ectopically expressed in PAO1 and cellular lysates were analysed by glycerol gradient fractionation. gp99 was detected by immunoblot and sedimented in ribosomal fractions. **c**, Western blot of ΦKZ-infected cells at indicated time points post-infection probed with a ΦKZ014 antibody. The asterisk (*) indicates antibody cross-reactivity with PAO1. Coomassie stain of the blot is shown as loading control. **d**, FLAG-tagged (ΦKZ014₋FLAG) was ectopically expressed in PAO1 and cellular lysates were analysed by glycerol gradient fractionation at either 1 mM (top) or 10 mM magnesium (bottom) and ΦKZ014₋FLAG was detected by immunoblotting. d.p. indicates a degradation product.

**e**, Anti-FLAG pull-down of ectopically expressed ΦKZ014₋FLAG from infected cells in conditions of 1 mM or 10 mM magnesium. Samples were eluted with FLAG peptide and analysed on SDS–PAGE gels (top). MS analysis of eluted samples and enrichment–abundance plot (bottom). **f**, ΦKZ014₋FLAG was ectopically expressed. Cellular lysates were treated with MNase or puromycin and analysed by sucrose gradient fractionation. ΦKZ014₋FLAG was detected by immunoblot. **g**, *P. aeruginosa* strain PAO1 was infected with ΦKZ and ΔΦKZ014 (Extended Data Fig. 5b,c). At 20 min post-infection, cells were lysed and complexes analysed in a sucrose gradient. Sedimentation of native ΦKZ014 was probed by immunoblot with an ΦKZ014 antibody. A₂₆₀ absorption profile indicates fractions that contain 70S and polysomes. M, size marker. **h**, 70S and polysomes were concentrated from the indicated fraction in **g** (bars) and analysed by western blotting. Ctrl, control; NI, non-infected; PI, post-infection.

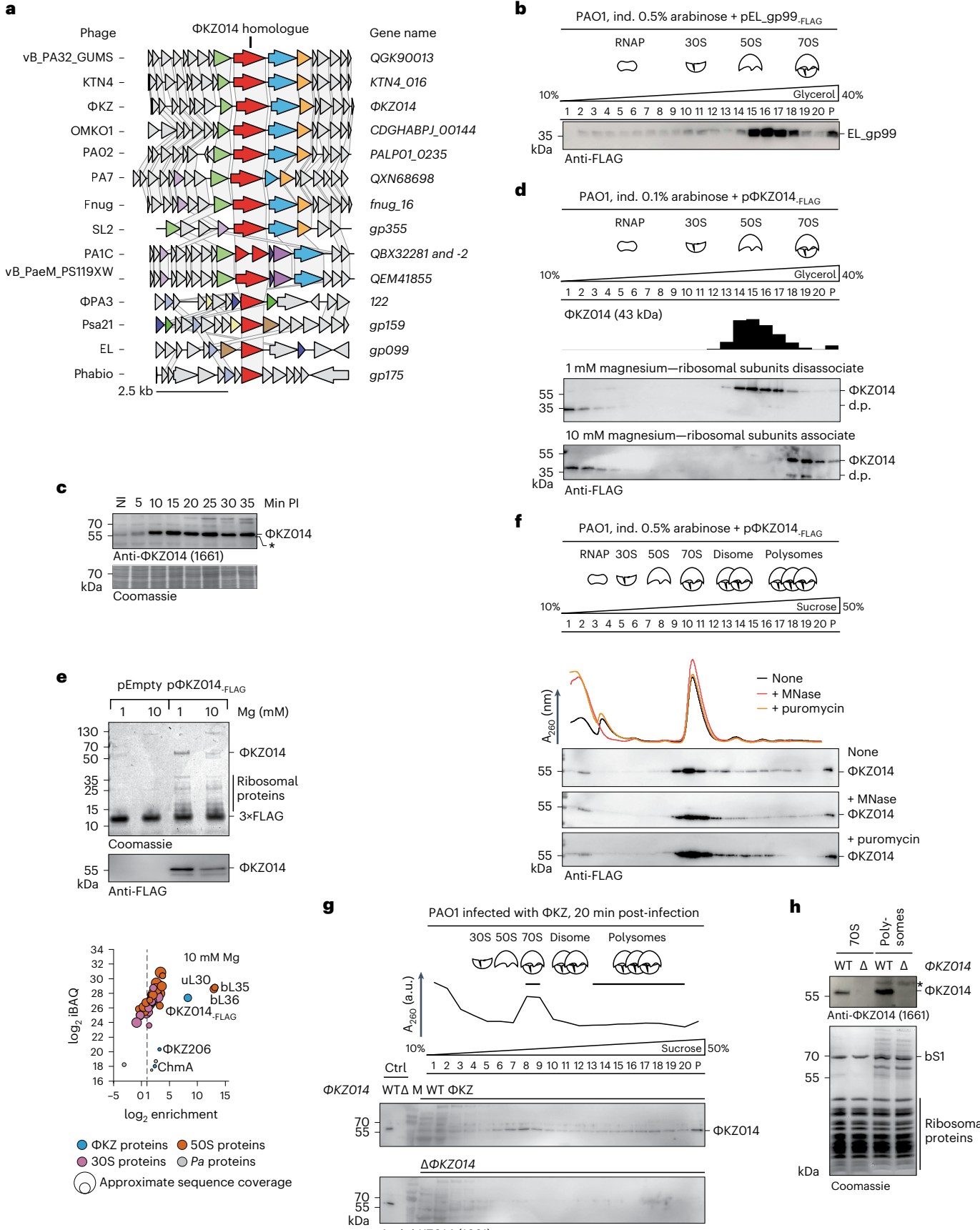

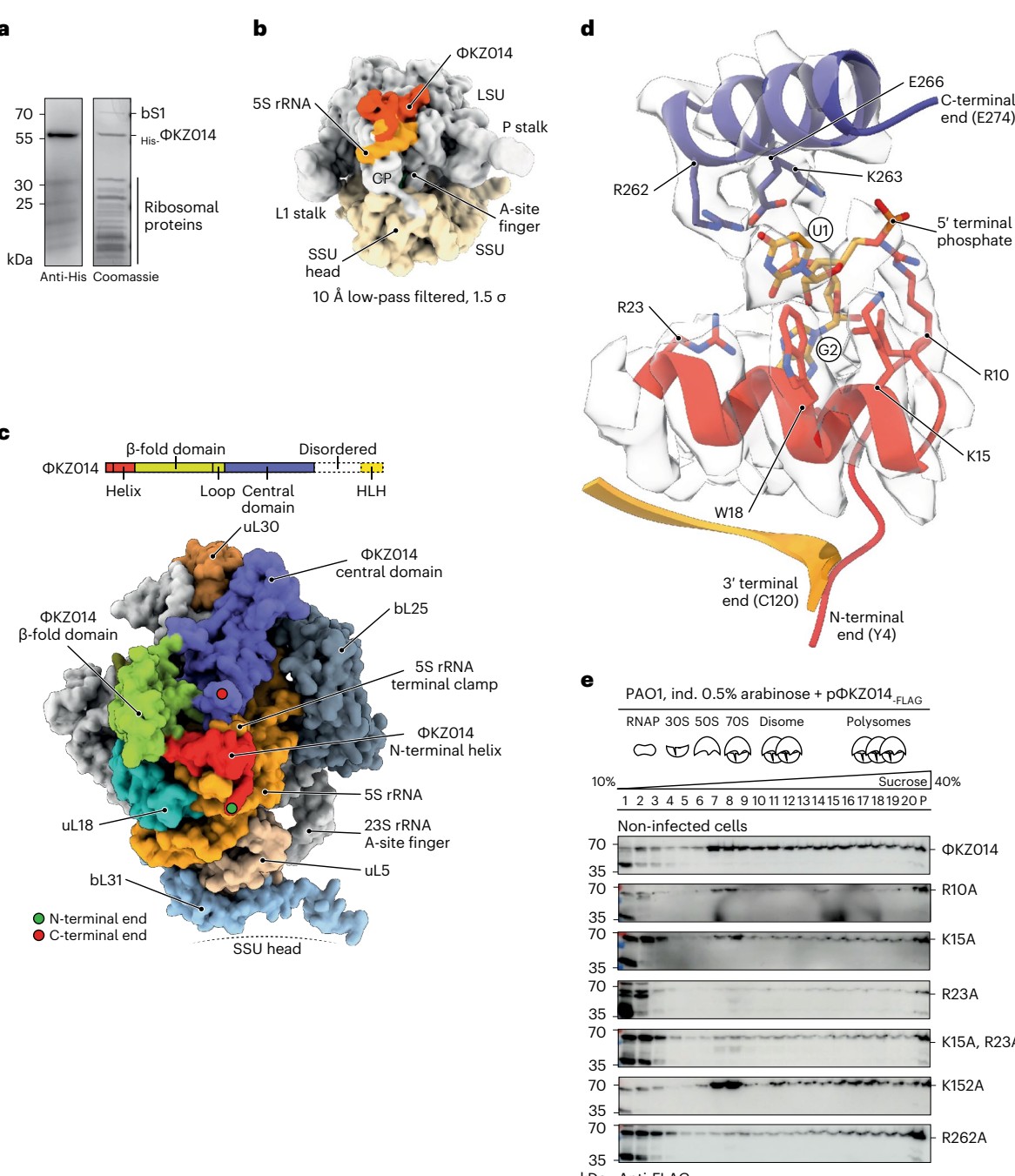

**Fig. 5 | ΦKZ014 clamps the 5S rRNA. a**, His-tagged ΦKZ014 (His-ΦKZ014) was ectopically expressed in *P. aeruginosa* and purified by immobilized metal-affinity chromatography. The protein composition of the eluted fraction was analysed by SDS–PAGE. **b**, Cryo-EM reconstruction of the density map of the 70S–ΦKZ014 complex (10 Å low-pass filtered). The ΦKZ014 density occupies the large ribosomal subunit on top of the 5S rRNA. **c**, Domain representation of ΦKZ014 (top). Surface representation of the 70S–ΦKZ014 model (bottom). ΦKZ014 binds at the 50S central protuberance (CP). The N-terminal helix of ΦKZ014 and the last helix of the central domain clamp the terminal end of the 5S rRNA. Both domains are connected by the β-fold domain. The C-terminal end is unstructured

and not resolved in the map. **d**, Terminal U1 of 5S rRNA is coordinated by ΦKZ014 residues R10, K15, W18, R262, K263 and E266. U1 is flipped out of the helical configuration. The second 5S residue G2 (base pairing with 5S residue C118) is coordinated by ΦKZ014 residue R23 at the base. The 3′-terminal end of 5S rRNA and the N-terminal end of ΦKZ014 are shown as cartoons without density. **e**, Key interacting residues K15, R23, K152 and R262A were exchanged in ΦKZ014-FLAG for alanine and the protein was expressed in *Pseudomonas*. Cellular lysates were analysed by sucrose gradient fractionation and ΦKZ014 variants were detected by immunoblotting. ΦKZ014 WT reference from Fig. 3d. HLH, helix–loop–helix; SSU, small ribosomal subunit; LSU, large ribosomal subunit.

biogenesis[26], we still lack a clear mechanistic understanding of how 5S rRNA contributes to protein synthesis. Targeted perturbation of translation by phage proteins could improve our mechanistic understanding of protein synthesis and the role of 5S rRNA therein.

ΦKZ014 is one of the earliest phage proteins expressed upon infection and it is important for phage propagation in certain *Pseudomonas*

strains. There are different ways in which it might function; for example, by directly modulating translation to ensure optimal translation of phage transcripts or by hindering a bacterial stress response that is activated at or targets the ribosome. Given that during ΦKZ infection, transcription and translation are spatially separated by the phage nucleus, one may envision a coupling of both processes

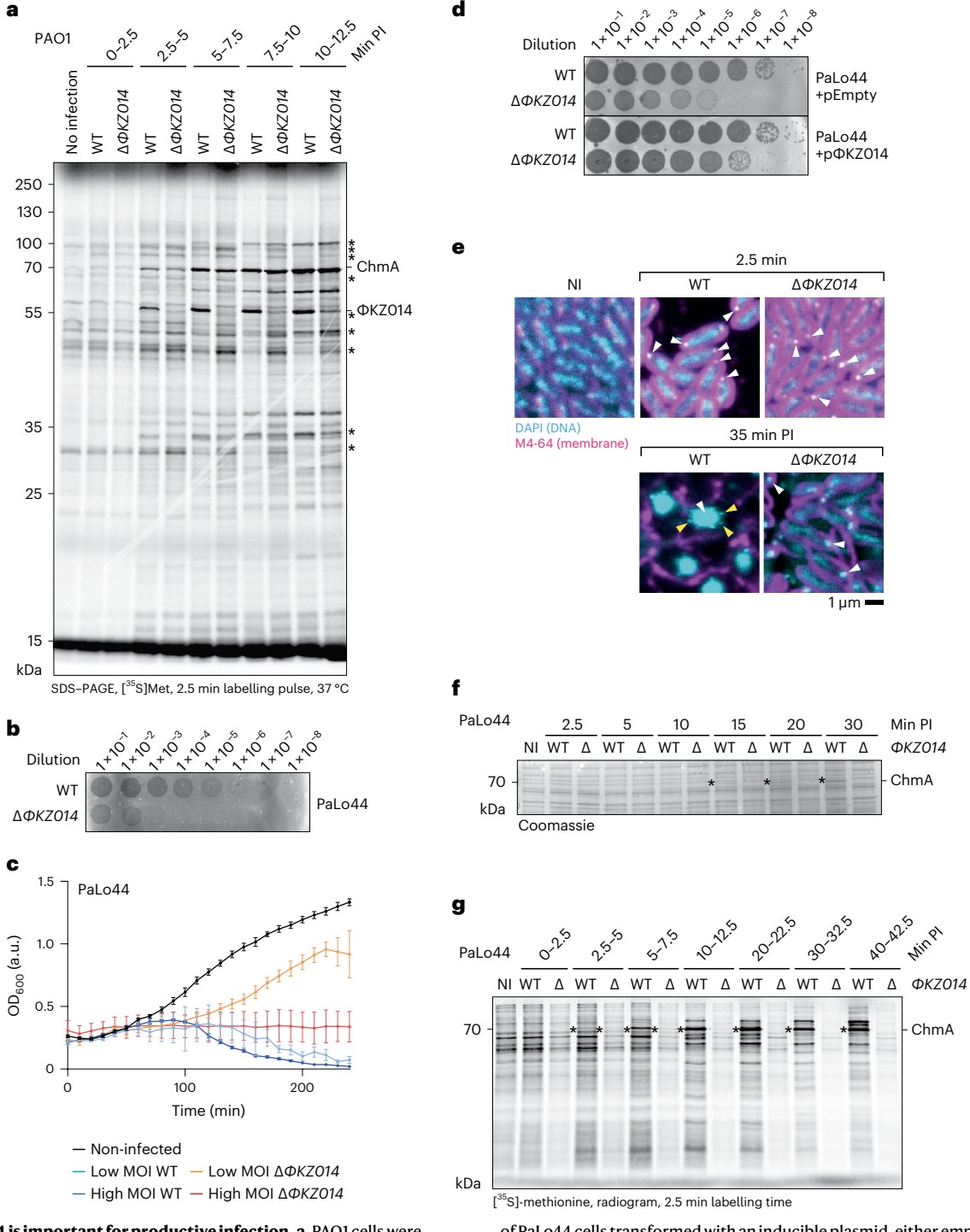

**Fig. 6 | ΦKZ014 is important for productive infection. a**, PAO1 cells were grown in minimal media and infected with ΦKZ. At the indicated time points, [35S]methionine was added and incorporated into newly translated proteins. The labelling pulse was quenched after 2.5 min by addition of excess unlabelled methionine and chloramphenicol. The protein composition in cells was analysed in an SDS–PAGE gel and 35S-labelled proteins were imaged by an autoradiogram. **b**, Plaque assay of the *Pseudomonas* clinical isolate PaLo44 infected with WT ΦKZ and the *ΦKZO14*-deleted phage. **c**, Lysis curve of PaLo44 cells infected with either WT ΦKZ or the *ΦKZO14*-deleted phage at a high or low MOI as indicated. The data represent the mean of three different cell clones, error bars indicate s.d. The results are representative of two independent experiments. **d**, Plaque assay

of PaLo44 cells transformed with an inducible plasmid, either empty or encoding for *ΦKZO14*. ΦKZO14 expression was induced with 0.5% arabinose. **e**, Fluorescence images of infected cells collected and chemically cross-linked at the indicated time points. DNA is stained with 4′,6-diamidino-2-phenylindole (DAPI; cyan) and membranes are stained with FM4-64 (magenta). Scale bar, 1 µm. White arrows point to the injected phage genome or phage nucleus; yellow arrows point to newly produced capsids that attach to the phage nucleus and are loaded with phage genomic DNA. **f**, PaLo44 cells were infected with ΦKZ, lysed at the indicated time points and analysed by SDS–PAGE and Coomassie staining. **g**, Protein synthesis was pulse radiolabelled as in **a** for ΦKZ-infected PaLo44 cells. The asterisks (*) in **a**, **f** and **g** indicate the ChmA protein.

on translocation of the phage mRNA to the cytosol. Notably, ΦKZO14 plateaus at ~2,000 copies per cell, fewer than the number of cellular ribosomes[33]. Hence, only a fraction of ribosomes will be bound by

ΦKZO14. It is tempting to speculate that ΦKZO14 might localize a subset of ribosomes to the vicinity of the early phage infection vesicle[22] or modify these ribosomes for efficient phage mRNA translation at this

site, but further investigations are required to test these hypotheses. The ΦKZ014 protein might also play a role in accelerating production of factors that neutralize host defence systems[18]. Phages and bacteria engage in an arms race that drives diversification and evolution of new host defence and phage takeover systems[34,35]. Notably, the stimuli that trigger defence systems remain poorly understood. Upon phage infection, many stress pathways are induced that can lead to abortive infection[36]. The translating ribosome represents a platform for surveillance[37] and it is conceivable that phages might have developed mechanisms to circumvent translation stress. Therefore, ΦKZ014 might act to overcome first-line defence mechanisms that are triggered at the ribosome. If it fails, this might lead to abortive infection, in line with the residual MOI-dependent ability of the ΦKZ014-deletion phage to promote cell lysis in several clinical isolates (Fig. 6c).

Overall our study highlights that phage genomes represent an important resource for the discovery of proteins that modulate the host gene expression machinery. Detailed investigation of the mechanisms of these phage proteins might lead to biotechnological or medical applications, for example, in vitro translation systems or inhibitors to recall phages used in phage therapy.

## Methods

### Phage lysate preparation

Bacterial cells were grown to an optical density at 600 nm ($OD_{600}$) of ~0.3 at 37 °C in LB media. Cells were infected with phage lysate (high titre $1$–$3 \times 10^{11}$ plaque-forming units (PFU)) at an MOI of between 0.1 and 1. In the following 3–4 h, lysis was observed and the solution was cleared at 5,000$g$ for 15 min at 4 °C. Phages were precipitated with one volume precipitation buffer (2.5 M NaCl and 20% (w/v) PEG 8000) for 1 h at 4 °C, and pelleted at 4,000$g$ for 10 min at 4 °C. Precipitation buffer was carefully removed and the pellet was resuspended in 0.5 volume phage storage buffer (50 mM Tris-HCl, pH 7.5, 100 mM NaCl and 16 mM $MgCl_2$). Lysates were stored at 4 °C and the phage titre was regularly validated by a plaque assay on PAO1.

### Phage plaque assay

Bacterial cells were grown to an $OD_{600}$ of ~0.3 at 37 °C. Soft agar (0.3–0.5% agar, 1 mM $MgCl_2$) was boiled and allowed to cool down to 42 °C. A 100 µl aliquot of bacterial cells was added to 10 ml soft agar, poured into a petri dish plate and solidified with a closed lid. The phage lysate was serially diluted in LB media from $1 \times 10^{-1}$ to $1 \times 10^{-8}$. A 5 µl volume of each phage lysate dilution was spotted onto the soft-agar plate. Overnight, plaques became visible and were counted to determine PFU per millilitre phage lysate.

### Glycerol gradient fractionation

*P. aeruginosa* PAO1 was grown to an $OD_{600}$ of ~0.3 at 37 °C and was infected with ΦKZ at an MOI of 15, as previously described[20]. Phage infection was allowed for 5 min without shaking and then cells were collected at 10 min, 20 min and 30 min post-infection on ice. Cells were pelleted and the different time points were pooled to extend our view on complexes in the first half of the replication cycle. A cell pellet from 180 ml infected cells ($OD_{600}$ ~ 0.3) was resuspended in 800 µl lysis buffer (20 mM Tris-HCl, pH 7.4, 150 mM KCl, 2.5 mM $MgCl_2$ and 1 mM dithiothreitol (DTT), 0.1% (v/v) Triton X-100 and 1 mM phenylmethylsulfonyl fluoride (PMSF)). Cells were lysed in 2 ml FastPrep tubes with lysing matrix E (MP Biomedicals) for 2× 40 s in a FastPrep-24 instrument (MP Biomedicals) at 6 m s⁻¹ at 4 °C. The lysate was cleared at 13,000$g$ for 10 min and layered onto a 10–40% glycerol gradient in lysis buffer without PMSF. The gradient was centrifuged for 17 h at 100,000$g$ and 4 °C. Twenty 590 µl fractions were collected and the remaining volume was used to resuspend the pellet extensively, representing the pellet fraction. Samples were mixed with 0.25 volume 4× Bolt sodium dodecyl sulfate (SDS) sample buffer (Invitrogen) and were boiled for 5 min at 95 °C. A 2 pmol human albumin sample was spiked-in per 50 µl fraction

for subsequent normalization. MS sample preparation and measurement was conducted at the Proteomics Core Facility EMBL Heidelberg by Jennifer Schwarz. Samples were reduced and alkylated with 10 mM DTT at 56 °C for 30 min and 2-chloroacetamide at room temperature in the dark for 30 min, respectively. Samples were cleaned up using the SP3 protocol[38]. Sequencing-grade Trypsin (300 ng; Promega) in 50 mM ammonium bicarbonate was added for overnight digestion at 37 °C. Peptides were recovered by collecting the supernatant on a magnet followed by a second elution with ultrapure water. Samples were dried under vacuum centrifugation and reconstituted in 10 µl 1% formic acid and 4% acetonitrile and then stored at −80 °C until liquid chromatography–MS analysis.

### MS analysis

Samples were analysed at the EMBL Proteomics Core Facility. An Ulti-Mate 3000 RSLC nano LC system (Dionex) fitted with a trapping cartridge (µ-Precolumn C18 PepMap 100, 5 µm particle size, 300 µm inside diameter × 5 mm length, 100 Å) and an analytical column (nanoEase M/Z HSS T3 column 75 µm × 250 mm C18, 1.8 µm, 100 Å; Waters) was coupled directly to a Q Exactive Plus (Thermo Scientific) mass spectrometer using the Nanospray Flex ion source in positive ion mode. Trapping was carried out with a constant flow of 0.05% trifluoroacetic acid at 30 µl min⁻¹ onto the trapping column for 4 min. Subsequently, peptides were eluted via the analytical column with a constant flow of 0.3 µl min⁻¹ with an increasing percentage of solvent B (0.1% formic acid in acetonitrile) from 2% to 4% in 4 min, from 4% to 8% in 2 min, from 8% to 25% for a further 89 min, from 25% to 40% in another 17 min, and finally from 40% to 80% in 3 min. The peptides were introduced into the Q Exactive Plus via a Pico-Tip Emitter (360 µm outside diameter × 20 µm inside diameter, 10 µm tip; MS Wil) and an applied spray voltage of 2.2 kV. The capillary temperature was set at 275 °C. Full mass scans were acquired with a mass range of 350–1,400 $m/z$ in profile mode with resolution of 70,000. The filling time was set at a maximum of 20 ms with a limitation of $3 \times 10^6$ ions. Data-dependent acquisition was performed with the resolution of the Orbitrap set to 17,500, with a fill time of 50 ms and a limitation of $1 \times 10^5$ ions. A normalized collision energy of 26 was applied; the loop count was 20; the isolation window was 1.7 $m/z$. A dynamic exclusion time of 30 s was used. The peptide match algorithm was set to 'preferred' and charge exclusion to 'unassigned'; charge states 1 and 5–8 were excluded. Tandem MS data were acquired in centroid mode. The raw MS data were processed with MaxQuant (v.1.6.17.0)[39] and searched against the UniProt databases UP000002438 and UP000002098 for *P. aeruginosa* and ΦKZ phage, respectively. As an internal standard, the entry P02768 (albumin of *Homo sapiens*) was used in each experiment. Common contaminants were included in each search. Decoy mode was set to revert. Carbamidomethyl I was set as fixed modification, acetylation of N termini and oxidation of methionine were set as variable modifications. The mass error tolerance for the full scan MS spectra was set to 20 ppm and for the tandem MS spectra to 0.5 Da. A maximum of two missed cleavages was permitted. For protein identification, a minimum of 1 unique peptide with a peptide length of at least 7 amino acids and a false discovery rate below 0.01 were required on the peptide and protein level. Match between runs was enabled with standard settings. Quantification was performed using intensities and iBAQ values[40], calculated as the sum of the intensities of the identified peptides and divided by the number of observable peptides of a protein. Intensities were normalized to the spike-in and the gradient fractions to obtain sedimentation profiles. t-Distributed stochastic neighbour embedding was conducted in Orange[41].

### Sequencing of RNA

PAO1 was grown to an $OD_{600}$ of 0.3 in LB media. Cells were washed and inoculated 1:3 in M9 minimal media and grown to an $OD_{600}$ of 0.5. Cells in SM buffer (50 mM Tris-HCl, pH 7.5, 100 mM NaCl and 8 mM $MgSO_4$) were infected with ΦKZ phage at an MOI of 15. Cells were lysed

with 1% SDS. The lysate was acidified with 0.3 M NaOAc, pH 5.2, and RNA was extracted with PCI and chloroform, and precipitated with ethanol/0.3 M NaOAc, pH 6.5. RNA was DNase digested. A 100 ng RNA sample was used for rRNA depletion with the RiboCop rRNA Depletion Kit for Mixed Bacterial Samples (META; Lexogen). RNA quality was assessed on a 2100 Bioanalyzer with the RNA 6000 Pico Kit (Agilent Technologies). Libraries were prepared with the CORALL RNA-Seq Library Prep Kit (Lexogen). PCR amplification was conducted for 14 cycles. Libraries were pooled and spiked with 1% PhiX control library. Sequencing was performed at 5 million reads per sample in single-end mode with 150 nt read length on the NextSeq 500 platform (Illumina) using a Mid Output sequencing kit. FASTQ files were demultiplexes with bcl2fastq2 v.2.20.0.422 (Illumina). Reads were trimmed with cutadapt (1.15 (ref. 42)). Reads were mapped to genomics sequences NC_002516 (PAO1) and NC_004629 (ΦKZ) and quantified with READemption 0.4.3 (ref. 43).

### Expression of phage proteins

PAO1 was transformed as previously described[44] with expression plasmids for arabinose-inducible C-terminally TEV–3×FLAG-tagged phage proteins in the backbone of pJN105. Briefly, an overnight culture of PAO1 was inoculated from overnight culture to an $OD_{600}$ of 0.05 in LB media (with 50 μg ml$^{-1}$ gentamicin), induced with 0.1–1% arabinose and grown to an $OD_{600}$ of 0.3. Cells were infected with high-titre ΦKZ phage (MOI of 15; $1–3 × 10^{11}$ PFU ml$^{-1}$), incubated for 5 min at room temperature, and then at 37 °C at 220 rpm for 5–20 min. The infection was stopped by rapid cool down on ice.

### Purification of ΦKZ014 protein and absolute quantification

ΦKZ014 purification under native conditions resulted in strong contamination with nucleic acids. Application of stringent conditions resulted in loss of the protein. For absolute quantification, which did not require folded protein, protein was purified under denaturing conditions for normalization in western blot detection. H6–3C–ΦKZ014 was expressed from pMiG-41 in PAO1. Cells were grown to an $OD_{600}$ of 0.6 in LB with 50 μg ml$^{-1}$ gentamicin at 37 °C and expression was induced with 0.5% arabinose for 2 h. Cells were collected and lysed by sonication for 2× 2 min (50% amplitude, Sonopuls HD 3200, VS70T tip; Bandelin) in lysis buffer (Tris-HCl, pH 8.0, 0.5 M KCl, 2 mM MgCl$_2$, 5 mM DTT and 1 mM PMSF). The lysate was cleared at 100,000$g$ for 1 h in a Ti70 rotor (Beckman Coulter). The pellet was subsequently resuspended in lysis buffer with 8 M urea and 20 mM imidazole at room temperature for 1 h. Non-solubilized protein was removed by a second pelleting at 100,000$g$ for 1 h in a Ti70 rotor (Beckman Coulter). The solubilized and denatured protein was loaded onto a 5 ml HisTrap HP column (Cytiva), washed with 6 M urea in lysis buffer and eluted with 350 mM imidazole in 6 M urea lysis buffer. ΦKZ014 was concentrated by centrifugal filtration (3 kDa cut-off, Amicon Ultra; Merck-Millipore). The protein concentration of the purified ΦKZ014 was estimated by correlation of band thickness in Coomassie staining with BSA at ~50 kDa in SDS–PAGE. Native ΦKZ014 in infected cells was detected by western blot (1:10,000 anti-ΦKZ014, catalogue number 1661; Eurogentec; 1:10,000 anti-rabbit-HRP, catalogue number 31460; Thermo Scientific) and cross-correlated to the signal of the purified protein. The cell number used as input for the western blot was determined by counting colonies and resulted in 2 × 10$^8$ colony forming units per ml at an $OD_{600}$ of ~0.5, 1/40 of a 50 ml culture was loaded per lane, hence 2.5 × 10$^8$ cells. The number of ribosomes per cell was determined by lysis of a defined number of cells, MNase treatment and gradient fractionation, followed by estimation of 70S ribosomes by the extinction coefficient 3.84 × 10$^7$ M$^{-1}$ cm$^{-1}$, which resulted in 4,000 ribosomes per cell at an $OD_{600}$ of ~0.3.

### Polysome profile fractionation

A total of 10–30 OD cells were resuspended on ice in 800 μl polysome lysis buffer (20 mM Tris-HCl, pH 7.4, 150 mM KCl, 10–70 mM MgCl$_2$, 2 mM DTT, 1 mM PMSF and 100 μg ml$^{-1}$ lysozyme) and 500 μl of 0.1 mM glass beads were added. Lysis was performed in the Retch M200 mill at 30 Hz for 5 min at 4 °C. The supernatant was cleared at 13,000$g$ for 10 min and layered onto a 10–50% (w/v) sucrose gradient in polysome lysis buffer without PMSF and lysozyme. The gradient was centrifuged for 16 h at 70,500$g$ and 4 °C. The polysome absorption profiles at 260 nm ($A_{260}$) were recorded on a BioComp fractionator. Twenty fractions (590 μl each) were collected and the pellet was resuspended in the last one and taken completely.

### SDS–PAGE

Samples were resuspended in SDS–PAGE loading dye (60 mM Tris-HCl, pH 6.8, 0.2 g ml$^{-1}$ SDS, 0.1 mg ml$^{-1}$ bromophenol blue, 77 mg ml$^{-1}$ DTT and 10% (v/v) glycerol), boiled and stored at 4 °C. Proteins were resolved on 12–15% acrylamide (37.5:1) gels in a discontinuous Laemmli buffer system constituting the separation gel (375 mM Tris-HCl, pH 8.8, 0.1% (w/v) SDS) and a stacking gel (125 mM Tris-HCl, pH 6.8, 0.1% (w/v) SDS), which were polymerized with 0.1% (w/v) ammonium persulfate and 0.001% (v/v) N,N,N′,N′-tetramethylethane-1,2-diamine (TEMED), and resolved in the running buffer (25 mM Tris-HCl, 192 mM glycine and 0.1% (w/v) SDS).

### Immunoblotting

SDS–PAGE gels were semi-dry blotted onto methanol-preactivated polyvinylidene membranes for 1 h at 34 mA per gel (20× 8 cm) in transfer buffer (200 mM Tris base, 150 mM glycine, 0.1% (w/v) SDS and 20% (v/v) methanol). Membranes were blocked with 5% BSA in TBS-T buffer (20 mM Tris-HCl, pH 7.4, 150 mM NaCl and 0.05% Tween 20). The following antibodies were used: anti-FLAG (mouse, 1:3,000, F1804; Sigma), anti-His (mouse, 1:3,000, A7058; Sigma), anti-GFP (mouse, 1:1,000, 11814460001; Roche), anti-mouse-HRP (goat, 1:10,000, 31430; Thermo Scientific), anti-rabbit-HRP (goat, 1:10,000, 31460; Thermo Scientific), anti-ΦKZ014 (1660, rabbit, 1:10,000, produced against peptide TEYDRNHGWNIREKH; Eurogentec; Extended Data Fig. 4d) and anti-ΦKZ014 (1661, rabbit, 1:10,000, produced against peptide EQYGESDDTSDESSY; Eurogentec; Extended Data Fig. 4d).

### Comparative gene cluster analysis

Gene homologues of ΦKZ014 were identified by BLAST (NCBI), JackHmmer (EMBL-EBI) and ColabFold AlphaFold multiple-sequence-alignment analysis. From indicated genes, the loci were obtained from NCBI with additional 3 kb flanking regions. Sequences were submitted to the online comparative gene cluster analysis (https://cagecat.bioinformatics.nl/) that uses cLinker for cluster generation[45].

### Cryo-EM sample preparation

H6–3C–ΦKZ014 was expressed from pMiG-41 in PAO1. Cells were grown to an $OD_{600}$ of 0.6 in LB with 50 μg ml$^{-1}$ gentamicin at 37 °C and expression was induced with 0.5% arabinose for 2 h. Cells were collected and lysed by sonication for 2× 2 min (50% amplitude, Sonopuls HD 3200, VS70T tip; Bandelin) in lysis buffer (20 mM Tris-HCl, pH 8.0, 500 mM KCl, 16 mM MgCl$_2$, 4 mM 2-ME, 1 mM PMSF and 20 mM imidazole). The lysate was cleared at 15,000$g$ for 20 min. Ribosomes in the supernatant were pelleted at 265,000$g$ for 1.5 h in a Ti70 rotor and resuspended in lysis buffer without PMSF. Equilibrated Protino Ni-IDA beads (Macherey-Nagel) were added and rotated for 30 min. Beads were washed 3× with 40 bead volumes lysis buffer. Complexes were eluted 2× with 300 mM imidazole in 1 ml lysis buffer. Eluted fractions were collected and layered onto 0.5 ml sucrose cushion (1.1 M sucrose) in grid buffer (20 mM HEPES-KOH, pH 7.5, 150 mM KCl, 16 mM MgCl$_2$, 1 mM DTT and 0.1% Triton X-100). Complexes were pelleted in a TLA110 rotor at 417,000$g$ for 1.5 h. The pellet was resuspended in grid buffer and adjusted to an $A_{260}$ of 7. Quantifoil grids (holey carbon R3/3 with a 2 nm carbon support) were glow-discharge treated at $10 × 10^{-1}$ Torr for 45 s. Samples were vitrified using a Vitrobot Mark IV (FEI), 3.5 μl was

applied, and after 45 s of incubation at 4 °C and 100% humidity, excess sample was blotted for 6 s and the grids were plunged into liquid ethane at approximately −160 °C.

## Cryo-EM imaging and reconstruction
Cryo-EM was carried out in the cryo-EM facility of the Julius-Maximilians University Würzburg. A total of 8,431 micrographs were collected on a Titan-Krios G3 with an X-FEG source and a Falcon III camera with direct electron detection at 300 kV, a magnification of ×75,000, a defocus range of −1.0 μm to −1.8 μm with 2.21 e⁻ per fraction and 40 fractions. The magnified pixel size was 1.0635 Å per pixel. All frames were aligned and summed using MotionCor2 (ref. [46]). Processing was performed with CryoSPARC v.3.3.2 (ref. [47]) (Extended Data Fig. 3b). Final maps were auto-sharpened in Phenix[48] for subsequent model building and structural refinement.

## Model building
The *Pseudomonas* 70S ribosome model was prepared by rigid-body docking of the reference model ribosomal subunits (PDB: 6SPG (ref. [49])), L31 was removed and homology modelled with Modeller 10.4 (ref. [50]) based on the *Escherichia coli* reference from PDB 7N1P (ref. [27]). tRNA was also incorporated from PDB 7N1P. 5S rRNA was modelled and refined in Coot 0.9.8.1 (refs. [51,52]), together with ΦKZ014, starting from AlphaFold2 and ColabFold[53,54] and Phenix 1.20.1-4487 (ref. [48]) map-to-model predictions. The final model was real-space refined and validated in Phenix. Structures and maps were displayed with UCSF ChimeraX (1.5 (ref. [55])).

## Phage engineering
ΦKZ genes were deleted or edited as described previously[56]. The editing plasmid template (pJG021) and the selection strain (PAO1 *tn7::Lse cas13a* transformed with pJG010 targeting *ΦKZ120* with guide RNA) were a kind gift from J. Bondy-Denomy (UCSF). The target gene was exchanged for the selection anti-CRISPR gene *acrVIA1*. We cloned an editing plasmid that was carrying ~500 bp up- and downstream regions of the target gene *ΦKZ014* and in between the *acrVIA1* gene (pMiG-068). *Pseudomonas* cells were infected with WT phage to allow for recombination, which generated a pool of phages also containing the edited phage. The edited phage was selected on the selection strain that was induced with 1 mM IPTG and 0.3% arabinose. From the resulting lysate, plaques were picked three times from an agar overlay with induced selection strain. The selected phage lysate was verified by PCR and whole genome sequencing for the loss of the target gene, incorporation of the *acrVIA1* gene and the correct locus.

Phage genomic DNA was prepared for long-read sequencing by Oxford Nanopore Technology (ONT) using the Rapid Barcoding Kit 24 v.14 (SQK-RBK114.24), following the manufacture's instructions. Sequencing was performed on a MinION flow cell (R10.4.1) for ~24 h in the accurate mode (260 bp). Base calling was performed with Guppy (v.6.3.8; ONT) in super-high-accuracy mode, and processing of the raw reads was conducted by Porechop (v.0.2.3; https://github.com/rrwick/Porechop). De novo assembly of the WT genome was carried out using the long-read assembler Flye (v.2.9.1 (ref. [57])) and ΔΦKZ014 mutant reads were subsequently mapped to the reference using minimap2 (v.2.17 (ref. [58])). The alignments were visually inspected using Integrative Genomics Viewer (IGV; v.2.16.1 (ref. [59])) software and no other alterations of the phage genome of ΔΦKZ014 beside *ΦKZ014::acrVIA1* were detected when compared with WT.

## Microscopy
Cells were grown to an OD₆₀₀ of 0.3 in LB media and infected with ΦKZ at an MOI of 5. After the indicated time points, 1 ml of cells was put on ice to quench infection progression. Cells were pelleted at 3,000g for 3 min at 4 °C. The pellet was resuspended in 4% paraformaldehyde in PBS and incubated for 15 min at 4 °C. Cells were washed in PBS,

resuspended in 50 μl PBS and stored overnight until microscopy. Cells were stained with 16 μM FM4-64 and 360 nM DAPI and layered onto 1.2% agarose pads. Transmission and fluorescence were detected with the confocal laser scanning microscope Leica SP5. Images were processed with ImageJ (1.53).

## Pulse labelling of translated proteins
PAO1 was grown overnight in M9 minimal media (12.8 g K₂HPO₄•7H₂O, 3 g KH₂PO₄, 0.5 g NaCl, 1 g NH₄Cl per 1 l, 0.4% glucose, 0.1 mM CaCl₂ and 2 mM MgSO₄ (ref. [60])). Cells were inoculated 1:100 in fresh media and grown to an OD₆₀₀ of ~0.3 at 37 °C, which represented an exponential growth state. Cells were infected with phage at an MOI of 4 in SM buffer. At defined time points, 10 μCi ml⁻¹ [³⁵S]methionine (Hartmann Analytic) were added for labelling of synthesized proteins. After 2.5 min, the reaction was stopped by addition of 50 μg ml⁻¹ methionine and 100 μg ml⁻¹ chloramphenicol and cells were put on ice. Cells were pelleted at 11,000g for 5 min at 4 °C, resuspended in 50 μl SDS–PAGE loading dye, boiled and resolved on a 15% SDS–PAGE gel. The gel was dried and autoradiograms were prepared.

## Statistics and reproducibility
Grad-seq data are a representative example of two independent experiments. Representative experiments were repeated 2× (Figs. 3c,d (ΦKZ105/-206), 4b–e, 5e, and 6c,d,f, and Extended Data Figs. 1b, 2, 3a and 5a–c), 3× (Figs. 4f, 5a, and 6a,e) and 4× (Figs. 3c,d (ΦKZ014), 4g,h and 6b,g) with similar results. No statistical method was used to predetermine sample size. No data were excluded from the analyses and the experiments were not randomized. In addition, the investigators were not blinded to allocation during experiments and outcome assessment.

## Reporting summary
Further information on research design is available in the Nature Portfolio Reporting Summary linked to this article.

## Data availability
MS data are deposited at the ProteomeXchange consortium via the PRIDE partner repository[61] with the data set identifier PXD038771 for ΦKZ-infected cells. Raw data after MaxQuant and sequencing analysis are listed in Supplementary Table 1. Sedimentation data can also be viewed in a user-friendly browser at www.helmholtz-hiri.de/en/datasets/gradseqphage. Raw sequencing data and coverage files are accessible at Gene Expression Omnibus[62] with the accession number GSE223979; the analysed data are listed in Supplementary Table 2. Cryo-EM density maps of 70S-tRNA(P)–ΦKZ014, 70S–ΦKZ014 (focussed) and 70S-tRNA(E)–ΦKZ014 were deposited at EMDB under accession number EMD-16566 (ref. [63]). The final model of 70S-tRNA(P)–ΦKZ014 was deposited at RCSB PDB 8CD1. Strains, oligonucleotides, plasmids, antibodies and software are listed in Supplementary Table 4. Source data are provided with this paper.

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

## Acknowledgements

We thank L. Vogel for technical assistance, A. Sparmann for discussions and for editing the article, N. Famelis for cloning of plasmids, E. Hauschild for microscopy, and A. Schlosser and S. Lamer from the technology platform MS (Würzburg University) and J. Schwarz from the EMBL Proteomics Core Facility (Heidelberg) for helpful discussions and MS. Cryo-EM was carried out in the cryo-EM facility of the Julius-Maximilians University Würzburg (DFG INST 93/903-1). We thank C. Makbul, J. Kuper and C. Kraft from the cryo-EM facility (University Würzburg) for helpful discussions. We thank the Core Unit Systems Medicine at the University of Würzburg for excellent technical support and RNA-seq data generation. This work was supported by the Interdisciplinary Center for Clinical Research (IZKF) Würzburg (project Z-06). We thank L. Putzeys for phage genome sequencing analysis and J. Bondy-Denomy (UCSF) for providing plasmids. L.W. holds a predoctoral scholarship from FWO-fundamental research (11D8920N). The article was supported by funding from the European Research Council under the European Union's Horizon 2020 Research and Innovation program (grant agreement number 819800) awarded to R.L. The work was funded by Deutsche Forschungsgemeinschaft (DFG) project 465133664 in the Priority Programme 'New Concepts in Prokaryotic Virus–Host Interactions – From Single Cells to Microbial Communities' (SPP 2330) awarded to J.V.

## Author contributions

M.G. and J.V. conceived, designed and interpreted the experiments, and wrote the manuscript. M.G. conducted the experiments and analysed the data. M.G. and K.C. constructed plasmids. L.W. screened clinical isolates and growth curves. J.V., R.L. and B.B. contributed critical resources and advice. All authors contributed to the article.

## Funding

## Competing interests

The authors declare no competing interests.

## Additional information

**Extended data** is available for this paper at https://doi.org/10.1038/s41564-024-01616-x.

**Correspondence and requests for materials** should be addressed to Jörg Vogel.

**a**

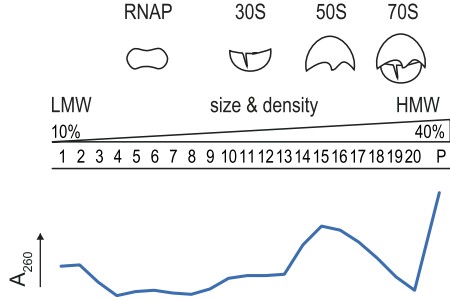

**b**

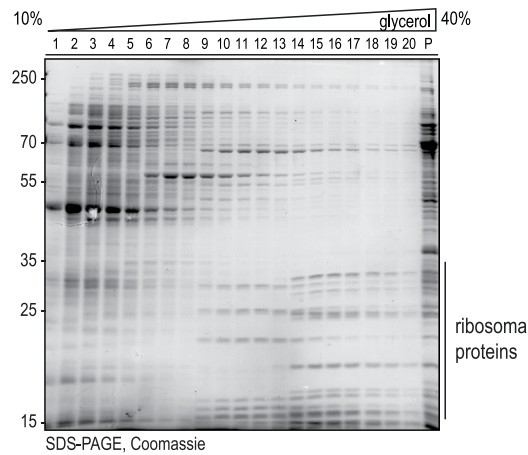

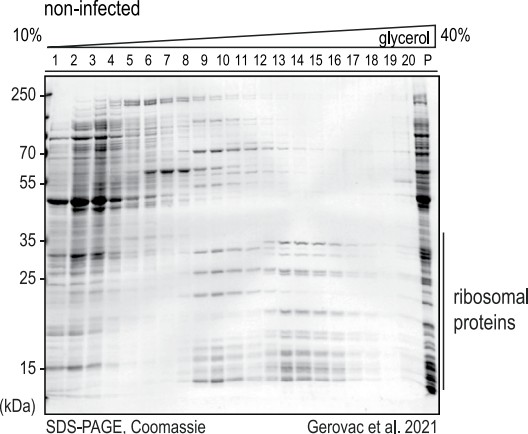

**c**

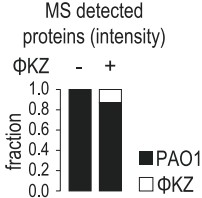

**Extended Data Fig. 1 | Integrity of the cellular proteome after infection is preserved. a**, Peaks in the absorption profile ($A_{260}$) of the cellular lysate indicate sedimentation positions of ribosomal subunits. **b**, The apparent protein pattern by size is not substantially changed upon infection (non-infected sample, reproduced from a previous publication (Gerovac et al.[20]). **c**, Relative mass-spectrometric quantification of protein abundance by iBAQ.

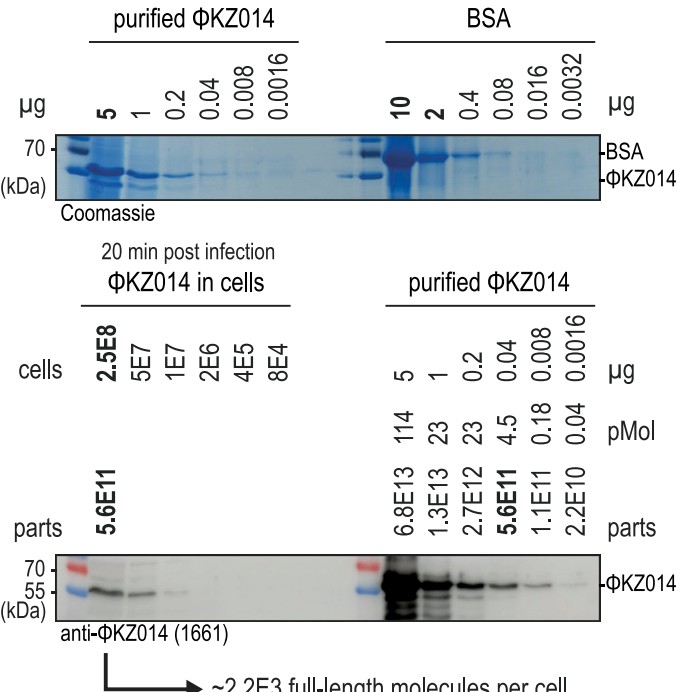

**Extended Data Fig. 2 | ΦKZ014 reaches 20-min post infection about two thousand copies per cell.** (Top) $_{His-}$ΦKZ014 was purified from PAO1 at denaturing conditions by immobilised metal affinity chromatography and a serial dilution was analysed on an SDS-PAGE and Coomassie staining. To obtain the amount of protein per dilution, bovine serum albumin (BSA) was serially diluted and used for cross-correlation. (Bottom) ΦKZ infected cells were lysed 20 min post-infection and analysed by SDS-PAGE. ΦKZ014 was detected by immunoblotting via a peptide specific antibody against a ΦKZ014 epitope. The same $_{His-}$ΦKZ014 purified samples from top were used to correlate the signal to protein amount per sample and cell counts based on colony forming units (CFU) per ml cell input.

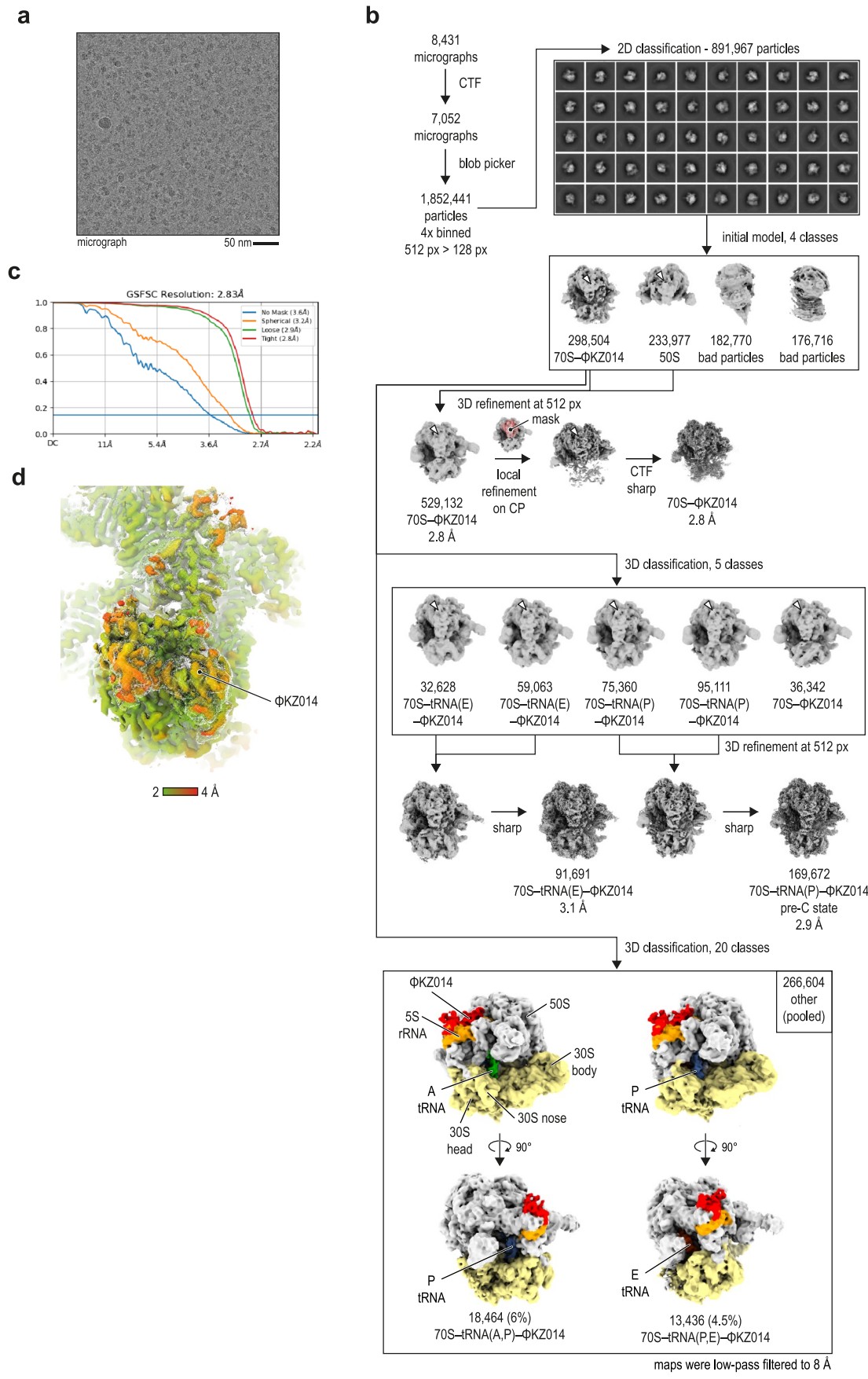

**Extended Data Fig. 3 | See next page for caption.**

**Extended Data Fig. 3 | ΦKZ014 occupies translating ribosomes. a**, Micrograph of the purified 70S–ΦKZ014 complex. **b**, In cryo-EM processing, we selected 50S– and 70S–ΦKZ014 3D classes and performed a focused refinement on ΦKZ014 extra density (red mask). In an additional path, the 70S–ΦKZ014 ribosome 3D class was further 3D classified into five classes yielding classes with or without densities for tRNAs in the P or E sites that were merged and refined to maps for 70S–tRNA(P)–ΦKZ014 and 70S–tRNA(E)–ΦKZ014 complexes. More extensive 3D classification with 20 classes revealed 70S–tRNA(A,P)–ΦKZ014 and 70S– tRNA(P,E)–ΦKZ014 classes. White arrow heads point to the extra density of ΦKZ014. tRNA positions relate to the 50S subunit. **c**, FSC curve, determined in CryoSPARC refinement. **d**, Local resolution estimation of the additional density that was attributed to ΦKZ014, determined in CryoSPARC.

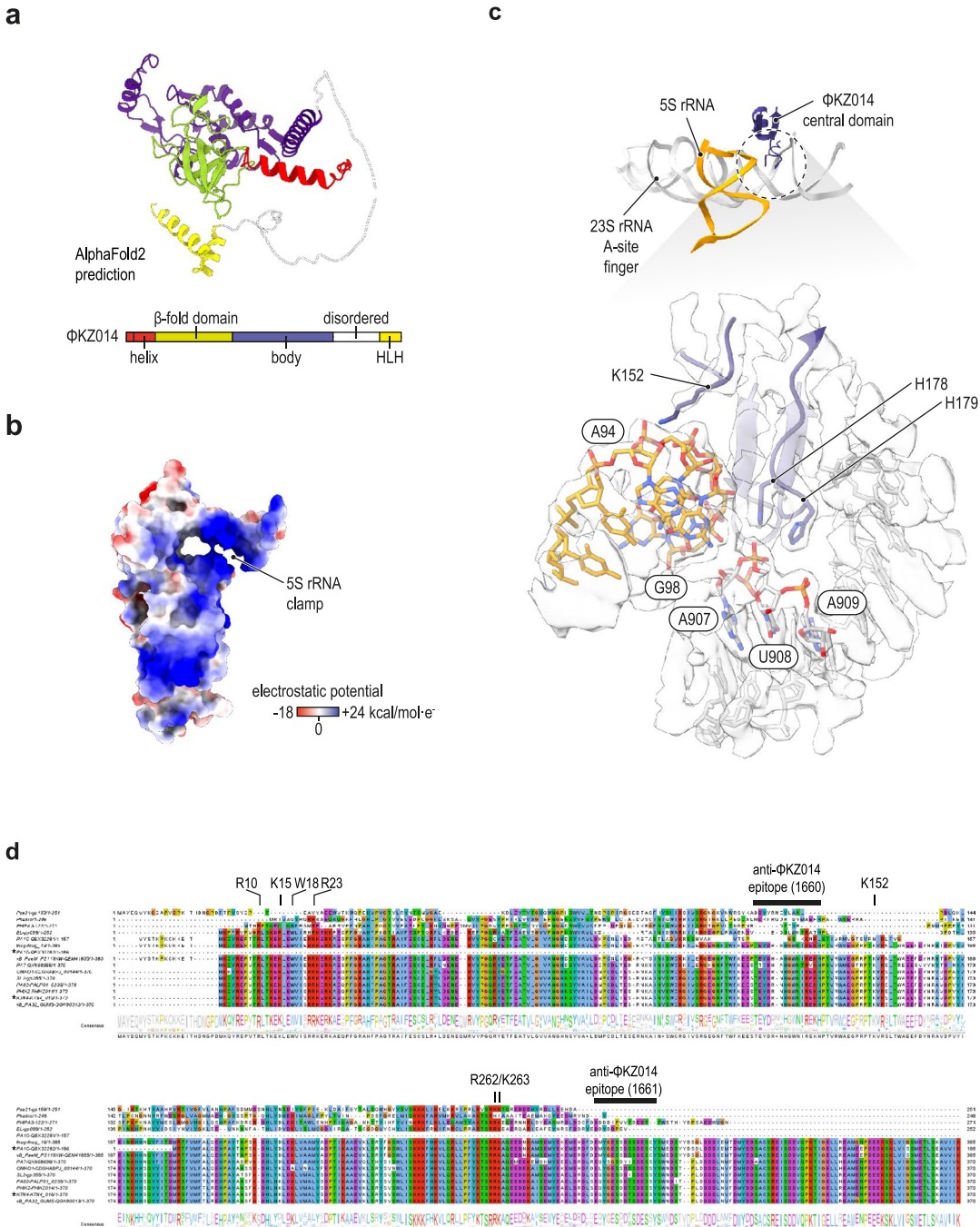

**Extended Data Fig. 4 | ΦKZ014 structure and interaction sites. a**, The predicted structure of ΦKZ014 determined with AlphaFold2/ColabFold suggested an N-terminal helix, a β-fold domain, a central domain, a disordered region and a C-terminal helix-loop-helix (HLH) domain. **b**, Electrostatic surface potential of ΦKZ014 displayed in the orientation from the 5S rRNA interface side.

The position of the 5S rRNA clamp is labelled. **c**, ΦKZ014 interacts with the 23S rRNA backbone at positions U908 and A909 via two sequential His178 and His179 residues. This is near 5S rRNA A94-G98. Lys152 points in the direction of 5S rRNA backbone. **d**, Key interacting residues (R10, K15, W18, R23, K152, R262, K263) are conserved in ΦKZ014 homologs.

**a**

empty plasmid

pΦKZ014

55 ⎯ ⎯ ΦKZ014

(kDa)

$^{35}$S-methionine
radiogram
SDS-PAGE

**b**

wt

ΔΦKZ014

1.5 ⎯ wt-locus
1.0 ⎯ Δ-locus
(kBp)

EtBr

**c**

wt

ΔΦKZ014

70 ⎯
55 ⎯ ⎯ ΦKZ014

anti-ΦKZ014 (1661)

55 ⎯

(kDa)

Coomassie

**d**

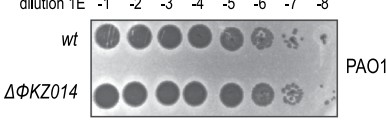

**e**

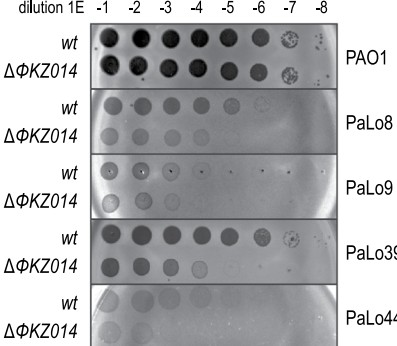

**Extended Data Fig. 5 | ΦKZ014 deletion in ΦKZ. a**, ΦKZ014 was overexpressed and translated proteins were pulse-radiolabelled with $^{35}$S-methionine. **b**, The genomic locus of *ΦKZ014* was amplified by PCR and resolved by agarose gel electrophoresis. In the *ΦKZ014*-deletion phage, the *ΦKZ014* coding sequence is exchanged for *acrVIA1*, resulting in a shorter product. **c**, *P. aeruginosa* PAO1 cells were infected with *wt* ΦKZ and ΔΦKZ014, harvested after 20 min and cellular lysates were analysed by SDS-PAGE. ΦKZ014 was detected by immunoblotting with a peptide-specific antibody against ΦKZ014. Coomassie staining served as loading control. **d**, Plaque forming assay using PAO1 cells infected with wt and ΔΦKZ014 phage. **e**, Extended Plaque forming assay with clinical isolate PaLo8/ -9/-39/-44 strains.

# Reporting Summary

## Statistics

For all statistical analyses, confirm that the following items are present in the figure legend, table legend, main text, or Methods section.

| n/a | Confirmed | |
|---|---|---|
| ☐ | ☒ | The exact sample size (*n*) for each experimental group/condition, given as a discrete number and unit of measurement |
| ☐ | ☒ | A statement on whether measurements were taken from distinct samples or whether the same sample was measured repeatedly |
| ☐ | ☒ | The statistical test(s) used AND whether they are one- or two-sided *Only common tests should be described solely by name; describe more complex techniques in the Methods section.* |
| ☒ | ☐ | A description of all covariates tested |
| ☒ | ☐ | A description of any assumptions or corrections, such as tests of normality and adjustment for multiple comparisons |
| ☐ | ☒ | A full description of the statistical parameters including central tendency (e.g. means) or other basic estimates (e.g. regression coefficient) AND variation (e.g. standard deviation) or associated estimates of uncertainty (e.g. confidence intervals) |
| ☐ | ☒ | For null hypothesis testing, the test statistic (e.g. *F*, *t*, *r*) with confidence intervals, effect sizes, degrees of freedom and *P* value noted *Give P values as exact values whenever suitable.* |
| ☒ | ☐ | For Bayesian analysis, information on the choice of priors and Markov chain Monte Carlo settings |
| ☒ | ☐ | For hierarchical and complex designs, identification of the appropriate level for tests and full reporting of outcomes |
| ☐ | ☒ | Estimates of effect sizes (e.g. Cohen's *d*, Pearson's *r*), indicating how they were calculated |

*Our web collection on statistics for biologists contains articles on many of the points above.*

## Software and code

Policy information about availability of computer code

| Data collection | no software was used for data collection |
|---|---|
| Data analysis | cutadapt (4.1), READemption (1.0.1), cLinker (0.0.24), MotionCor2 (1.4.2), CryoSPARC (3.3.2), AlphaFold2 (2), Colabfold - AlphaFold2_mmseqs (1.3.0), ChimeraX (1.4), Phenix (1.20.1-4487), WinCoot (0.9.8.1), DelPhi (8.5.0), MaxQuant (v1.6.17.0), Porechop (v0.2.3), Flye (v2.9.1), minimap2 (v2.17), IGV (v2.16.1), Grad-seq browser (1.0) |

For manuscripts utilizing custom algorithms or software that are central to the research but not yet described in published literature, software must be made available to editors and reviewers. We strongly encourage code deposition in a community repository (e.g. GitHub). See the Nature Portfolio guidelines for submitting code & software for further information.

## Data

Policy information about availability of data

All manuscripts must include a data availability statement. This statement should provide the following information, where applicable:
- Accession codes, unique identifiers, or web links for publicly available datasets
- A description of any restrictions on data availability
- For clinical datasets or third party data, please ensure that the statement adheres to our policy

MS data are deposited at the ProteomeXchange consortium via the PRIDE partner repository (Perez-Riverol et al., 2022) with the data set identifier PXD038771 for ΦKZ infected cells. Raw data after MaxQuant and sequencing analysis are listed in Suppl. Table 1. Sedimentation data can also be viewed in a user-friendly browser

at www.helmholtz-hiri.de/en/datasets/gradseqphage. Raw sequencing data and coverage files are accessible at Gene Expression Omnibus (Barrett et al., 2012) with the accession number GSE223979, the analysed data are listed in Suppl. Table 2. Cryo-EM density maps of 70S–tRNA(P)–ΦKZ014, 70S–ΦKZ014 (focused), 70S–tRNA(E)–ΦKZ014 were deposited at EMDB under accession number EMD-16566. The final model of 70S–tRNA(P)–ΦKZ014 was deposited at RCSB-PDB 8CD1. Strains, oligonucleotides, plasmids, antibodies, software are listed in Suppl. Table 4.

## Research involving human participants, their data, or biological material

Policy information about studies with [human participants or human data](). See also policy information about [sex, gender (identity/presentation), and sexual orientation]() and [race, ethnicity and racism]().

| | |
|---|---|
| Reporting on sex and gender | n/a |
| Reporting on race, ethnicity, or other socially relevant groupings | n/a |
| Population characteristics | n/a |
| Recruitment | n/a |
| Ethics oversight | n/a |

Note that full information on the approval of the study protocol must also be provided in the manuscript.

# Field-specific reporting

Please select the one below that is the best fit for your research. If you are not sure, read the appropriate sections before making your selection.

☒ Life sciences  ☐ Behavioural & social sciences  ☐ Ecological, evolutionary & environmental sciences

For a reference copy of the document with all sections, see [nature.com/documents/nr-reporting-summary-flat.pdf](http://nature.com/documents/nr-reporting-summary-flat.pdf)

# Life sciences study design

All studies must disclose on these points even when the disclosure is negative.

| | |
|---|---|
| Sample size | No statistical method was used to predetermine sample size. Bacterial assays were completed in two independent experiments and the error accounts were reported. All attempts to replicate data were successful. Grad-seq data are a representative example of two independent experiments. |
| Data exclusions | We did not observe outliers in our data that needed to be excluded. |
| Replication | Bacterial assays were completed in two independent experiments and the error accounts were reported. All attempts to replicate data were successful. Grad-seq data are a representative example of two independent experiments. Representative experiments were repeated two (Figs. 3c,d(ΦKZ105/-206), 4b-e, 5e, 6c,d,f, Ext. Data Figs. 1b, 2, 3a, 5a-c), three (Figs. 4f, 5a, 6a,e), four times (Figs. 3c,d(ΦKZ014), 4g,h, 6b,g) with similar results. We added a "Statistics and Reproducibility" section in the methods part. |
| Randomization | No data were excluded from the analyses and the experiments were not randomized. |
| Blinding | This study does not involve procedures that require blinding. The investigators were not blinded to allocation during experiments and outcome assessment. |

# Reporting for specific materials, systems and methods

We require information from authors about some types of materials, experimental systems and methods used in many studies. Here, indicate whether each material, system or method listed is relevant to your study. If you are not sure if a list item applies to your research, read the appropriate section before selecting a response.

## Materials & experimental systems

| n/a | Involved in the study |
|-----|----------------------|
| ☐ | ☒ Antibodies |
| ☒ | ☐ Eukaryotic cell lines |
| ☒ | ☐ Palaeontology and archaeology |
| ☒ | ☐ Animals and other organisms |
| ☒ | ☐ Clinical data |
| ☒ | ☐ Dual use research of concern |
| ☒ | ☐ Plants |

## Methods

| n/a | Involved in the study |
|-----|----------------------|
| ☒ | ☐ ChIP-seq |
| ☒ | ☐ Flow cytometry |
| ☒ | ☐ MRI-based neuroimaging |

## Antibodies

| | |
|---|---|
| Antibodies used | anti-FLAG (mouse, 1:3k, Sigma, F1804), anti-His (mouse, 1:3k, Sigma, A7058), anti-GFP (mouse, 1:1k, Sigma, 11814460001 (Roche)), anti-mouse-HRP (goat, 1:10k, Thermo Scientific, 31430), anti-rabbit-HRP (goat, 1:10k, Thermo Scientific, 31460), anti-ΦKZ014.1 (1660, rabbit, 1:10k, Eurogentec, produced against peptide EQYGESDDTSDESSY, Ext. Data 4d). |
| Validation | For anti-FLAG, anti-His, anti-GFP, anti-mouse and anti-rabbit, see the product information sheets of the manufacturers (Sigma, Thermo Scientific), these were also validated in previous publications from the Vogel lab with various ectopically expressed targets (e.g. Gerovac et al. 2020 RNA). anti-ΦKZ014 antibodies were produced by Eurogentec and validated in Fig. 4c,g,h and Extended Data 2 with uninfected vs infected PAO1 cells by PHIKZ phage, PHIKZ014 deleted strains, and with purified PHIKZ014 protein. |

