## [Peer Review File · Nature Microbiology]

Peer Review Information

Journal: Nature Microbiology

Manuscript Title: Phage proteins target and co-opt host ribosomes immediately upon infection

Corresponding author name(s): Professor Jörg Vogel

Editorial Notes:

This manuscript has been previously reviewed at another journal. This document only contains reviewer comments, rebuttal and decision letters for versions considered at Nature Microbiology. Mentions of prior referee reports have been redacted.

Reviewer Comments & Decisions:Decision Letter, initial version:**Message:** 20th November 2023

Dear Professor Vogel,

Thank you for your patience while your manuscript "Immediate targeting of host ribosomes by jumbo phage-encoded proteins" was under peer-review at Nature Microbiology. It has now been seen by two of the original referees, whose expertise and comments you will find at the of this email. I should note that we did not send this back to the original Reviewer #3, as they did not have major concerns last time. You will see from their comments below that, in general, they find the work improved, however there are still some important points which should be addressed with a revision. We remain very interested in the possibility of publishing your study in Nature Microbiology, but would like to consider your response to these concerns in the form of a revised manuscript before we make a final decision on publication. In particular, Reviewer #1 points out a number of editorial concerns, as well as a request to see a clearer blot in figure 4. I would like to note again that, as per our previous discussions, we still will not require a more flushed out mechanism to meet our editorial bar for publication here--however, if you can provide more insight in that regard, we do agree that it would lead to a stronger paper!

If you have not done so already please begin to revise your manuscript so that it conforms to our Article format instructions at <http://www.nature.com/nmicrobiol/info/final-submission/>

The usual length limit for a Nature Microbiology Article is six display items (figures or tables) and 3,000 words. We have some flexibility, and can allow a revised manuscript at 3,500 words, but please consider this a firm upper limit. There is a trade-off of ~250 words per display item, so if you need more space, you could move a Figure or Table to Supplementary Information.

Some reduction could be achieved by focusing any introductory material and moving it to the start of your opening 'bold' paragraph, whose function is to outline the background to your work, describe in a sentence your new observations, and explain your main conclusions. The discussion should also be limited. Methods should be described in a separate section following the discussion, we do not place a word limit on Methods.

Nature Microbiology titles should give a sense of the main new findings of a manuscript, and should not contain punctuation. Please keep in mind that we strongly discourage active verbs in titles, and that they should ideally fit within 90 characters each (including spaces).

We strongly support public availability of data. Please place the data used in your paper into a public data repository, if one exists, or alternatively, present the data as Source

1Data or Supplementary Information. If data can only be shared on request, please explain why in your Data Availability Statement, and also in the correspondence with your editor. For some data types, deposition in a public repository is mandatory - more information on our data deposition policies and available repositories can be found at <https://www.nature.com/nature-research/editorial-policies/reporting-standards#availability-of-data>.

Please include a data availability statement as a separate section after Methods but before references, under the heading "Data Availability". This section should inform readers about the availability of the data used to support the conclusions of your study. This information includes accession codes to public repositories (data banks for protein, DNA or RNA sequences, microarray, proteomics data etc...), references to source data published alongside the paper, unique identifiers such as URLs to data repository entries, or data set DOIs, and any other statement about data availability. At a minimum, you should include the following statement: "The data that support the findings of this study are available from the corresponding author upon request", mentioning any restrictions on availability. If DOIs are provided, we also strongly encourage including these in the Reference list (authors, title, publisher (repository name), identifier, year). For more guidance on how to write this section please see: <http://www.nature.com/authors/policies/data/data-availability-statements-data-citations.pdf>

To improve the accessibility of your paper to readers from other research areas, please pay particular attention to the wording of the paper's opening bold paragraph, which serves both as an introduction and as a brief, non-technical summary in about 150 words. If, however, you require one or two extra sentences to explain your work clearly, please include them even if the paragraph is over-length as a result. The opening paragraph should not contain references. Because scientists from other sub-disciplines will be interested in your results and their implications, it is important to explain essential but specialised terms concisely. We suggest you show your summary paragraph to colleagues in other fields to uncover any problematic concepts.

If your paper is accepted for publication, we will edit your display items electronically so they conform to our house style and will reproduce clearly in print. If necessary, we will re-size figures to fit single or double column width. If your figures contain several parts, the parts should form a neat rectangle when assembled. Choosing the right electronic format at this stage will speed up the processing of your paper and give the best possible results in print. We would like the figures to be supplied as vector files - EPS, PDF, AI or postscript (PS) file formats (not raster or bitmap files), preferably generated with vector-graphics software (Adobe Illustrator for example). Please try to ensure that all figures are non-flattened and fully editable. All images should be at least 300 dpi resolution (when figures are scaled to approximately the size that they are to be printed at) and in RGB colour format. Please do not submit Jpeg or flattened TIFF files. Please see also 'Guidelines for Electronic Submission of Figures' at the end of this letter for further detail.

Figure legends must provide a brief description of the figure and the symbols used, within 350 words, including definitions of any error bars employed in the figures.

When submitting the revised version of your manuscript, please pay close attention to our [href="https://www.nature.com/nature-research/editorial-policies/image-integrity">Digital Image Integrity Guidelines](https://www.nature.com/nature-research/editorial-policies/image-integrity) and to the following points below:

-- that unprocessed scans are clearly labelled and match the gels and western blots presented in figures.

2-- that control panels for gels and western blots are appropriately described as loading on sample processing controls
-- all images in the paper are checked for duplication of panels and for splicing of gel lanes.

Please include a statement before the acknowledgements naming the author to whom correspondence and requests for materials should be addressed.

Finally, we require authors to include a statement of their individual contributions to the paper -- such as experimental work, project planning, data analysis, etc. -- immediately after the acknowledgements. The statement should be short, and refer to authors by their initials. For details please see the Authorship section of our joint Editorial policies at http://www.nature.com/authors/editorial_policies/authorship.html

* include a point-by-point response to any editorial suggestions and to our referees. Please include your response to the editorial suggestions in your cover letter, and please upload your response to the referees as a separate document.

* ensure it complies with our format requirements for Letters as set out in our guide to authors at www.nature.com/nmicrobiol/info/gta/

* state in a cover note the length of the text, methods and legends; the number of references; number and estimated final size of figures and tables

* resubmit electronically if possible using the link below to access your home page:

*This url links to your confidential homepage and associated information about manuscripts you may have submitted or be reviewing for us. If you wish to forward this e-mail to co-authors, please delete this link to your homepage first.

Please ensure that all correspondence is marked with your Nature Microbiology reference number in the subject line.

Nature Microbiology is committed to improving transparency in authorship. As part of our efforts in this direction, we are now requesting that all authors identified as 'corresponding author' on published papers create and link their Open Researcher and Contributor Identifier (ORCID) with their account on the Manuscript Tracking System (MTS), prior to acceptance. This applies to primary research papers only. ORCID helps the scientific community achieve unambiguous attribution of all scholarly contributions. You can create and link your ORCID from the home page of the MTS by clicking on 'Modify my Springer Nature account'. For more information please visit www.springernature.com/orcid.

We hope to receive your revised paper within three weeks. If you cannot send it within this

3time, please let us know.

Reviewers Comments:

Reviewer #1 (Remarks to the Author):

This paper has improved substantially through the revisions made. In particular, I think the expansion of Fig. 6 has added nicely to the paper and strongly hints at an Abi mechanism in PaLo44 cells that phiK104 helps the phage evade. There are also modest 'collateral' effects of having phiK104 (as now shown in Fig. 6a) in the absence of the Abi system, but these effects don't impact phage replication in any significant way, with respect to plaque formation. I really think it wouldn't be terribly difficult to pinpoint the Abi system responsible (experimentally, not computationally as tried) and it would enhance the impact of the paper, but I gather the authors and editor have agreed this isn't needed for publication. So I'm generally supportive of proceeding at this stage, but there are a couple of things I think the authors need to fix, and one experiment that sorely needs a cleaner version to enable its interpretability.

- I don't really understand the rationale stated early on that " Jumbo phages present particularly exciting opportunities to identify molecular factors crucial to a successful host take-over, since they feature extraordinarily large genomes (200-600 kb) that encode hundreds of proteins of currently unknown function." Even an RNA phage with 4 ORFs can successfully replicate, so it's not clear that a bigger genome is needed to take over a host.

" Phages require mechanisms that enable them to immediately seize control of the host gene expression machinery to efficiently proceed through the infection process. It is well established that they manipulate the host RNA polymerase (RNAP) and overload the translation machinery with phage-derived transcripts." The first sentence should be modified to indicate that some phages require such mechanisms - some, like RNA phages, don't. Similarly, only some phages manipulate RNAP - some, like T7, make their own RNAP. This is addressed a few sentences later but this sentence quoted here should still be made accurate.

line 80: should say 'RNA polymerases' not 'polymerases'

The blot in Fig. 4g is impossible to interpret - there are clear bands in fractions 8 and 9, corresponding to the 50S, but the other HMW lanes are spotty and in some cases there appear to be multiple possible bands. To support the claim in the text that phiKZ104 associates with ribosomes during infection, this blot needs to be improved and rendered 'cleaner' like those in Fig. 4f.

Reviewer #2 (Remarks to the Author):

Ultimately in the revised version the authors do not really address completely any of major deficiencies in the first version of the manuscript.

41. The authors validate that some of the identified phage proteins bind to ribosomes but the mechanism of action of the phage proteins are not revealed. The conclusion that it binds to ribosomes is solid. The knockout suggests that translation is affected but whether this is a direct effect or not remains somewhat unclear in the absence of some in vitro validation. I understand that the editors agreed that deciphering the mechanism is beyond the scope of the manuscript, however, I would argue that at least showing that the binding of the phage protein to the ribosome is critical for its function (and the observed effects) should be shown. Since the authors have mutants that prevent the phage protein from binding to the ribosome, one wonders whether introduction of these mutants generates strains that behave similarly to the knockout strains? i.e. whether ribosome binding is actually necessary for the function of these proteins. This is not conclusively demonstrated. The authors propose models that the phage protein might affect localization of the phage-protein bound ribosomes, at least some experimental evidence showing this would considerably strengthen the manuscript in the lack of other mechanistic insights.

2. The authors explain that the strain specificity may arise from differences in defense systems however unfortunately this is also left hanging in the manuscript.

Lastly, the authors argue that the major advance of the study is the application of Grad-seq to identify phage proteins that are likely to interact with host protein complexes. While I agree that this is an exciting first step to identify novel proteins that interact with ribosomes, the method of running sucrose gradients to identify factors that migrate with ribosomal complexes is not novel or a major advance and has been used for decades to look for novel factors that interact with ribosomes. Thus, while I appreciate that the amount of work that the authors have undertaken, I find that the identification of the potential proteins that interact with host cell machinery in the absence of any functional insights not a sufficient advance for Nature Microbiol but maybe the editors think otherwise.

Author Rebuttal to Initial comments

Reviewers Comments:

Reviewer #1 (Remarks to the Author):

This paper has improved substantially through the revisions made. In particular, I think the expansion of Fig. 6 has added nicely to the paper and strongly hints at an Abi mechanism in PaLo44 cells that phiK104 helps the phage evade. There are also modest 'collateral' effects of having phiK104 (as now shown in Fig. 6a) in the absence of the Abi system, but these effects don't impact phage replication in any significant way, with respect to plaque formation. I really think it wouldn't be terribly difficult to pinpoint the Abi system responsible (experimentally, not computationally as tried) and it would enhance the impact of the paper, but I gather the authors and editor have agreed this isn't needed for publication. So I'm generally supportive of

5proceeding at this stage, but there are a couple of things I think the authors need to fix, and one experiment that sorely needs a cleaner version to enable its interpretability.

We appreciate the encouraging words of the reviewer. We have been attempting to identify the putative defense system experimentally, but this proved more challenging than might have been anticipated. Initially, we undertook a transposon insertion screen in PaLo44 using the mini-Tn5 system (Martínez-García et al. 2014 Front Bioeng Biotechnol 2(46)), aiming to score restored replication of the $\Delta\Phi KZ014$ phage. Unfortunately, both transformation and conjugation efficiency are low in PaLo44, and therefore the genome was only partly covered. Moreover, in the selected antibiotic resistant strains, we found single nucleotide polymorphisms that allowed growth under antibiotic selection rather than transposon insertions. This is a common issue when working with clinical isolates of *Pseudomonas*.

As an alternative approach, we plan to clone fragments from the PaLo44 genome into the PAO1 strain to transfer the defense system. However, this approach comes with its own challenges, e.g., if the putative defense system requires additional genes outside of its own locus. Moreover, the defense system trigger might be dependent on the strain background or on intrinsic strain-specific stress levels. In this experimental set-up, positive selection for the desired phenotype is another challenge. If the defense system is transferred, cells will die upon infection. If not, the $\Delta\Phi KZ014$ phage will lyse the PAO1 cells. Therefore, selection must be conducted using a microcolony-based assay, in which colony survival is used as a positive selection criterium (Vassallo et al. 2022 Nat Microbiol 7(10), 1568-79). In these experimental settings, if the defense system eliminates the phage only one cell is killed, while the colony survives. The system could then still be identified in the surviving cells.

In summary, the experimental discovery of that putative abortive infection (Abi) or defense systems has turned out less straight-forward than it seems at first glance, and it will require us set up a genetic screen in a clinical isolate of *Pseudomonas* with poor genetic tractability. We are actively working on it, but this is turning into a project in its own right.

- I don't really understand the rationale stated early on that " Jumbo phages present particularly exciting opportunities to identify molecular factors crucial to a successful host take-over, since they feature extraordinarily large genomes (200-600 kb) that encode hundreds of proteins of currently unknown function." Even an RNA phage with 4 ORFs can successfully replicate, so it's not clear that a bigger genome is needed to take over a host.

Our intention was to argue that because jumbo phages have large genomes, there is ample opportunity to identify molecular factors with novel functions. But we appreciate the reviewers

point and changed the text accordingly: "... to identify molecular factors with novel functions for a successful host take-over..."

" Phages require mechanisms that enable them to immediately seize control of the host gene expression machinery to efficiently proceed through the infection process. It is well established that they manipulate the host RNA polymerase (RNAP) and overload the translation machinery with phage-derived transcripts." The first sentence should be modified to indicate that some phages require such mechanisms - some, like RNA phages, don't. Similarly, only some phages manipulate RNAP - some, like T7, make their own RNAP. This is addressed a few sentences later but this sentence quoted here should still be made accurate.

We changed this sentence accordingly (p. 2): "Most phages require mechanisms that enable them to immediately take control of gene expression in the infected cell to efficiently proceed through the replication process."

line 80: should say 'RNA polymerases' not 'polymerases'

This was changed accordingly.

The blot in Fig. 4g is impossible to interpret - there are clear bands in fractions 8 and 9, corresponding to the 50S, but the other HMW lanes are spotty and in some cases there appear to be multiple possible bands. To support the claim in the text that phiKZ104 associates with ribosomes during infection, this blot needs to be improved and rendered 'cleaner' like those in Fig. 4f.

This blot proved tricky, indeed, but we have repeated the experiment and now provide an improved blot in Fig. 4g. In this experiment we included the $\Delta\Phi KZ014$ strain as a control to discriminate between the specific $\Phi KZ014$ band and cross reactions of the antibody. We also show the 260 nm absorption profile of the gradient fractions, which enables better assignment of the 70S and polysome fractions (Fig. R1). In addition, in Fig. 4h, we concentrated the 70S and polysome fractions and loaded equal levels of ribosomes on a blot. We detected $\Phi KZ014$ in wt phages, but not the $\Delta\Phi KZ014$ strain, in both polysome and the 70S fractions, consistent with the claim that $\Phi KZ014$ occupies translating ribosomes. Of note, in Fig. 4f, we detected ectopically overexpressed $\Phi KZ014$ -FLAG by the FLAG-tag. In Fig. 4g, $\Phi KZ014$ was detected by a polyclonal antiserum raised against a single peptide of $\Phi KZ014$. This antibody is not as sensitive as the monoclonal anti-FLAG antibody and shows more cross-reactivity. This is also apparent from Fig. 4c.

a

b

Fig. R1 | Φ KZ014 sediments in polysome fractions. **a.** PAO1 was infected with Φ KZ and cellular lysates were analysed in a sucrose gradient 10-50%. Φ KZ014 was detected in a western blot by a peptide-specific antibody (anti- Φ KZ014, 1661). Non-infected and $\Delta\Phi$ KZ014-infected samples are included to visualize cross-reactions of the antibody. **b.** 70S and polysomes were concentrated from indicated fractions (a, black bars) and analysed by western blotting for Φ KZ014. * indicates cross-reactive bands.

Reviewer #2 (Remarks to the Author):

Ultimately in the revised version the authors do not really address completely any of major deficiencies in the first version of the manuscript.

1. The authors validate that some of the identified phage proteins bind to ribosomes but the mechanism of action of the phage proteins are not revealed. The conclusion that it binds to ribosomes is solid. The knockout suggests that translation is affected but whether this is a direct effect or not remains somewhat unclear in the absence of some in vitro validation. I understand that the editors agreed that deciphering the mechanism is beyond the scope of the manuscript, however, I would argue that at least showing that the binding of the phage protein to the ribosome is critical for

8

its function (and the observed effects) should be shown. Since the authors have mutants that prevent the phage protein from binding to the ribosome, one wonders whether introduction of these mutants generates strains that behave similarly to the knockout strains? i.e. whether ribosome binding is actually necessary for the function of these proteins. This is not conclusively demonstrated.

We attempted to mutate $\Phi KZ014$ in the phage genome, but this did not work in our hands. It seems that the phage genome does not permit integration of a selection cassette. *Chimalliviridae* genome manipulation is known to be a big challenge, as also evidenced in Guan et al. 2022 Nat Microbiol 7, 1956-66 (see, e.g., gene gp54 in Table 1), where manyattempts to tag a protein in the phage genome were unsuccessful. We also attempted to complement the $\Delta KZ014$ phage with $\Phi KZ014^{R23A}$ expressed in trans in the bacterial cell (as we did with wt $\Phi KZ014$ in Fig. 6d). However, in this experimental setting it is difficult to achieve the correct expression level. It is important to note that the $\Phi KZ014$ variants that we generated retain a limited ability to bind the ribosome. Especially under overexpression conditions, this might be sufficient to partially complement the phenotype. Given that our data indicate that all of the $\Phi KZ014$ protein is bound to the ribosome (Fig. 2a), we are convinced that this is its primary place of action.

The authors propose models that the phage protein might affect localization of the phage-protein bound ribosomes, at least some experimental evidence showing this would considerably strengthen the manuscript in the lack of other mechanistic insights.

The idea that $\Phi KZ014$ might affect the localization of ribosomes is based on accepted transcription-translation models and inspired by a recent cryo-ET analysis that showed that the early infection vesicle of ΦKZ is decorated with polysomes (Armbruster et al. 2023 bioRxiv, doi: 10.1101/2023.09.20.558163, Fig. 4c, Fig. 5c). Given that $\Phi KZ014$ is expressed very early upon infection and occupies translating ribosomes, we found it interesting and valid to speculate that $\Phi KZ014$ might localise a subset of ribosomes to the vicinity of the phage nucleus. The text clearly indicates that this is only a speculation, and we now included the statement that further research is needed to test these models. Generally, jumbo phage biology is a fast-moving field, and we think it is important to put our findings into context with the models emerging through the work of other labs. Yet, if this is still considered too speculative, we can delete this section of the discussion.

additional statement p. 9: “It is **tempting to speculate** that $\Phi KZ014$ might localise a subset of ribosomes to the vicinity of the phage nucleus or modify these ribosomes for efficient phage mRNA translation at this site (Armbruster et al. 2023) but further investigations are required to test this hypothesis.”

2. The authors explain that the strain specificity may arise from differences in defense systems however unfortunately this is also left hanging in the manuscript.

As discussed in more detail in our response to reviewer#1, it is technically challenging to experimentally identify the defense system that causes the strain specificity. The link to the defense system could be also indirect and related to a stress response at the translating ribosome as elaborated on in the discussion (p. 9). Again, we are working hard on developing a genetic system to identify that putative Abi or defense system.

Lastly, the authors argue that the major advance of the study is the application of Grad-seq to identify phage proteins that are likely to interact with host protein complexes. While I agree that this is an exciting first step to identify novel proteins that interact with ribosomes, the method of running sucrose gradients to identify factors that migrate with ribosomal complexes is not

10novel or a major advance and has been used for decades to look for novel factors that interact with ribosomes. Thus, while I appreciate that the amount of work that the authors have undertaken, I find that the identification of the potential proteins that interact with host cell machinery in the absence of any functional insights not a sufficient advance for Nature Microbiol but maybe the editors think otherwise.

We are afraid that there is a slight misunderstanding: Grad-seq does not use sucrose gradients to selectively identify proteins in ribosome fractions; instead, the technique uses glycerolgradients coupled to RNA-seq and proteomics to cover a much larger range of complexes between >200 kDa and 4 MDa (Smirnov et al. 2016 PNAS 113(41):11591-11596). This is what makes our study a unique resource for phage biology. It really is the first of its kind and establishes a framework for looking at new molecular interactions between the phage and its host in general. In this study, we focus on phage proteins that associate with host ribosomes because we deemed them particularly interesting and novel.

Of course, we are aware that gradients were historically used to identify proteins in ribosomal fractions (Gerovac, Vogel, Smirnov 2021 Front Mol Bio Sci, 8:661448). Our main claim is not related to the development of this method, but to the identification of phage factors that fractionate in high molecular weight fractions and therefore likely form part of cellular complexes. In this manuscript, we focused on ribosome associated phage factors. The existence of such factors had not been shown before and our observations open a new branch in the phage and ribosome fields. We validated our claim by biochemical and cryo-EM studies and show that one of the identified factors, Φ KZ014, has a functional role in translation (**Fig. 6a**). We are currently investigating the precise mechanism of action of Φ KZ014, but, as agreed with the editor, this is out of the scope of the current manuscript.

Decision Letter, first revision:

Message: Our ref: NMICROBIOL-23092308B

11th January 2024

Dear Dr. Vogel,

Thank you for your patience as we've prepared the guidelines for final submission of your Nature Microbiology manuscript, "Immediate targeting of host ribosomes by jumbo phage-encoded proteins" (NMICROBIOL-23092308B). Please carefully follow the step-by-step instructions provided in the attached file, and add a response in each row of the table to indicate the changes that you have made. Please also check and comment on any additional marked-up edits we have proposed within the text. Ensuring that each point is addressed will help to ensure that your revised manuscript can be swiftly handed over to our production team.

1If you have not done so already, please alert us to any related manuscripts from your group that are under consideration or in press at other journals, or are being written up for submission to other journals (see: <https://www.nature.com/nature-research/editorial-policies/plagiarism#policy-on-duplicate-publication> for details).

In recognition of the time and expertise our reviewers provide to Nature Microbiology's editorial process, we would like to formally acknowledge their contribution to the external peer review of your manuscript entitled "Immediate targeting of host ribosomes by jumbo phage-encoded proteins". For those reviewers who give their assent, we will be publishing their names alongside the published article.

Nature Microbiology offers a Transparent Peer Review option for new original research manuscripts submitted after December 1st, 2019. As part of this initiative, we encourage our authors to support increased transparency into the peer review process by agreeing to have the reviewer comments, author rebuttal letters, and editorial decision letters published as a Supplementary item. When you submit your final files please clearly state in your cover letter whether or not you would like to participate in this initiative. Please note that failure to state your preference will result in delays in accepting your manuscript for publication.

Cover suggestions

COVER ARTWORK: We welcome submissions of artwork for consideration for our cover. For more information, please see our [guide for cover artwork](https://www.nature.com/documents/Nature_covers_author_guide.pdf).

Nature Microbiology has now transitioned to a unified Rights Collection system which will allow our Author Services team to quickly and easily collect the rights and permissions required to publish your work. Approximately 10 days after your paper is formally accepted, you will receive an email in providing you with a link to complete the grant of rights. If your paper is eligible for Open Access, our Author Services team will also be in touch regarding any additional information that may be required to arrange payment for your article.

Please note that *Nature Microbiology* is a Transformative Journal (TJ). Authors may publish their research with us through the traditional subscription access route or make their paper immediately open access through payment of an article-processing charge (APC). Authors will not be required to make a final decision about access to their article until it has been accepted. [Find out more about Transformative Journals](https://www.springernature.com/gp/open-research/transformative-journals)

Authors may need to take specific actions to achieve

<https://www.springernature.com/gp/open-research/funding/policy-compliance-faqs> > **compliance with funder and institutional open access mandates.** If your research is supported by a funder that requires immediate open access (e.g. according to <https://www.springernature.com/gp/open-research/plan-s-compliance> > Plan S principles) then you should select the gold OA route, and we will direct you to the compliant route where possible. For authors selecting the subscription publication route, the journal's standard licensing terms will need to be accepted, including <https://www.nature.com/nature-portfolio/editorial-policies/self-archiving-and-license-to-publish> > self-archiving policies . Those licensing terms will supersede any other terms that the author or any third party may assert apply to any version of the manuscript.

For information regarding our different publishing models please see our <https://www.springernature.com/gp/open-research/transformative-journals> > Transformative Journals page. If you have any questions about costs, Open Access requirements, or our legal forms, please contact ASJournals@springernature.com.

Best regards,

Reviewer #1:

Remarks to the Author:

The final revisions, including the updated gel in Fig. 4, look fine so I support publication at this stage.

Final Decision Letter:

Message: 19th January 2024

Dear Jörg,

I am pleased to accept your Article "Phage proteins target and co-opt host ribosomes immediately upon infection" for publication in Nature Microbiology. Thank you for having chosen to submit your work to us and many congratulations.

Over the next few weeks, your paper will be copyedited to ensure that it conforms to

3Nature Microbiology style. We look particularly carefully at the titles of all papers to ensure that they are relatively brief and understandable.

Please note that *Nature Microbiology* is a Transformative Journal (TJ). Authors may publish their research with us through the traditional subscription access route or make their paper immediately open access through payment of an article-processing charge (APC). Authors will not be required to make a final decision about access to their article until it has been accepted. [Find out more about Transformative Journals](https://www.springernature.com/gp/open-research/transformative-journals)

Authors may need to take specific actions to achieve [compliance with funder and institutional open access mandates](https://www.springernature.com/gp/open-research/funding/policy-compliance-faqs). If your research is supported by a funder that requires immediate open access (e.g. according to [Plan S principles](https://www.springernature.com/gp/open-research/plan-s-compliance)) then you should select the gold OA route, and we will direct you to the compliant route where possible. For authors selecting the subscription publication route, the journal's standard licensing terms will need to be accepted, including [self-archiving and-](https://www.nature.com/nature-portfolio/editorial-policies/self-archiving-and-)

license-to-publish">self-archiving policies. Those licensing terms will supersede any other terms that the author or any third party may assert apply to any version of the manuscript.

With kind regards,